# Endothelial cell signature in muscle stem cells validated by VEGFA-FLT1-AKT1 axis promoting survival of muscle stem cell

**Mayank Verma**[1,2,3,4], **Yoko Asakura**[2,3,4], **Xuerui Wang**[2,3,4], **Kasey Zhou**[2,3,4], **Mahmut Ünverdi**[2,3,4], **Allison P Kann**[5,6], **Robert S Krauss**[5,6], **Atsushi Asakura**[2,3,4]*

[1]Department of Pediatrics & Neurology, Division of Pediatric Neurology, The University of Texas Southwestern Medical Center, Dallas, United States; [2]Stem Cell Institute, University of Minnesota Medical School, Minneapolis, United States; [3]Greg Marzolf Jr. Muscular Dystrophy Center, University of Minnesota Medical School, Minneapolis, United States; [4]Department of Neurology, University of Minnesota Medical School, Minneapolis, United States; [5]Department of Cell, Developmental, and Regenerative Biology, Icahn School of Medicine at Mount Sinai, New York, United States; [6]Graduate School of Biomedical Sciencesf, Icahn School of Medicine at Mount Sinai, New York, United States

**Abstract** Endothelial and skeletal muscle lineages arise from common embryonic progenitors. Despite their shared developmental origin, adult endothelial cells (ECs) and muscle stem cells (MuSCs; satellite cells) have been thought to possess distinct gene signatures and signaling pathways. Here, we shift this paradigm by uncovering how adult MuSC behavior is affected by the expression of a subset of EC transcripts. We used several computational analyses including single-cell RNA-seq (scRNA-seq) to show that MuSCs express low levels of canonical EC markers in mice. We demonstrate that MuSC survival is regulated by one such prototypic endothelial signaling pathway (VEGFA-FLT1). Using pharmacological and genetic gain- and loss-of-function studies, we identify the FLT1-AKT1 axis as the key effector underlying VEGFA-mediated regulation of MuSC survival. All together, our data support that the VEGFA-FLT1-AKT1 pathway promotes MuSC survival during muscle regeneration, and highlights how the minor expression of select transcripts is sufficient for affecting cell behavior.

*For correspondence:
asakura@umn.edu

**Competing interest:** The authors declare that no competing interests exist.

## Editor's evaluation

This study presents a valuable finding on the unique role of VEGFA-FLT1-AKT1 signaling in regulating muscle stem cell (MuSC) survival. The evidence supporting the claims is convincing, with multiple approaches utilized, including pharmacological and genetic methods performed in vitro and in vivo, demonstrating that the VEGFA-FLT1-AKT1 axis protected MuSCs from apoptosis. The work will be of broad interest to researchers in the MuSC biology field and support the future development of VEGFA and FLT1 targeted therapies for various diseases, such as cancer and neuromuscular diseases.

## Introduction

Skeletal muscle and endothelial cells (ECs) and their progenitors from the trunk and limbs are derived from the somites during early developments. Previous works demonstrated the existence of bipotent progenitors which express both *Pax3* and *Flk1* (*Eichmann et al., 1993*; *Kardon et al., 2002*;

*Ema et al., 2006*; *Esner et al., 2006*; *Tozer et al., 2007*). These bipotent progenitors migrate into trunk and limb buds from ventrolateral region of the somites to generate MYOD(+) myogenic cells followed by skeletal muscle and PECAM1(+) ECs followed by vasculatures (*Hutcheson and Kardon, 2009*; *Kardon et al., 2002*; *Lagha et al., 2009*; *Mayeuf-Louchart et al., 2016*; *Mayeuf-Louchart et al., 2014*). In addition, FLK1(+) cells give rise to myogenic cells during development and oncologic transformation (*Drummond and Hatley, 2018*; *Mayeuf-Louchart et al., 2014*; *Motoike et al., 2003*). Lastly, multipotent mesoangioblasts, vessel-associated stem cells, have been identified in embryonic dorsal aorta (*Minasi et al., 2002*). These cells are able to differentiate into several types of mesodermal tissues including skeletal muscle and ECs (*Roobrouck et al., 2011*). Interestingly, these myogenic cells show the same morphology as muscle satellite cells (MuSCs), stem cell populations for skeletal muscle, and express a number of myogenic and EC markers such as *Myod*, *Cdh15*, *Kdr*, and *Cdh5* (*De Angelis et al., 1999*). However, it is not clear whether adult MuSCs derived from these bipotent progenitors still maintain canonical EC signals. Curiously, blood vessel-associated myoendothelial cell progenitors that express both myogenic and EC markers, and are able to differentiate into myogenic cells following transplantation have been identified in the interstitial spaces of both murine and human adult skeletal muscle (*Tamaki et al., 2002*; *Zheng et al., 2007*; *Huang et al., 2014*). However, the relationship between these myoendothelial cell progenitors and MuSCs remains unclear.

Vascular endothelial growth factor (VEGF), specifically VEGFA modulates many biological aspects including angiogenesis through its two receptors, FLT1 and FLK1. Although FLK1 possesses stronger signaling capability and the major signaling receptor tyrosine kinase (RTK) for VEGFA, FLT1 has considerably higher affinity for VEGF but weaker cytoplasmic signaling capability. In normal tissue, FLT1 acts as a decoy receptor and a sink trap for VEGF thereby preventing excessive normal and pathological angiogenesis. In addition, there are two co-receptors for VEGFA (NRP1 and NRP2) that function with FLK1 to modulate VEGFA signaling. While VEGF signaling has been extensively studied for its role in development, proliferation, and survival of endothelial cells (ECs), its role in non-vascular systems such as neuron and bone has only recently been appreciated (*Poesen et al., 2008*; *Liu et al., 2012*; *Okabe et al., 2014*). Skeletal muscle tissue is the most abundant producer of VEGFA in the body. It has already been extensively studied in the skeletal muscle fibers in models of *Vegfa* knockout mice (*Tang et al., 2004*; *Wagner et al., 2006*; *Olfert et al., 2009*) as well as *Vegfa* overexpression (*Arsic et al., 2004*; *Yan et al., 2005*; *Messina et al., 2007*; *Bouchentouf et al., 2008*).

Adult skeletal muscle also contains the tissue resident muscle stem cell population, termed MuSCs, which mediate postnatal muscle growth and muscle regeneration (*Motohashi and Asakura, 2014*). After muscle injury, quiescent MuSCs initiate proliferation to produce myogenic precursor cells, or myoblasts. The myoblasts undergo multiple rounds of cell division before terminal differentiation and formation of multinucleated myotubes by cell fusion. Importantly, the MuSC-derived myoblasts also express VEGFA, which has been shown to increase the proliferation of myoblasts (*Christov et al., 2007*). Our data obtained from genetical model mice demonstrated that MuSCs express abundant VEGFA, which recruits ECs to establish vascular niche for MuSC self-renewal and maintenance (*Verma et al., 2018*). In addition, VEGFA and its receptors are expressed in the myoblast cell line, C2C12 cells, and the signaling can induce cell migration and protect apoptotic cell during myogenic differentiation in vitro (*Germani et al., 2003*; *Bryan et al., 2008*; *Mercatelli et al., 2010*). However, it is not clear whether MuSCs also express VEGF receptors and if cell-autonomous VEGFA signaling plays an essential roles in MuSC function during muscle regeneration in vivo.

We have previously shown that *Flt1* heterozygous gene knockout and conditional deletion of *Flt1* in ECs display increased capillary density in skeletal muscle, indicating the essential roles for *Flt1* in adult skeletal muscle. More importantly, when crossed with the Duchenne muscular dystrophy (DMD) model *mdx* mice, these mice show both histological and functional improvements of their dystrophic phenotypes. This was partly due to the effect of increased ECs leading to an increase in MuSCs (*Verma et al., 2010*; *Verma et al., 2019*; *Bosco et al., 2021*). However, the effect of VEGFA on MuSC in vivo remained unknown. We found that MuSCs express low levels of canonical EC markers including VEGF receptors using single cell transcriptomics. Therefore, we examined the effects of VEGFA on MuSCs and show that it has a drastic effect on cell survival in the via its receptor FLT1 by signaling through AKT1.

# Results

## EC gene signal including *Vegf receptors* in MuSCs

EC signatures in MuSCs has been seen in several gene expression data sets (*Figure 1—figure supplement 1A–1D*, *Supplementary file 1*; *Fukada et al., 2007*; *Charville et al., 2015*; *Ryall et al., 2015*; *van Velthoven et al., 2017*). However, with the lack of EC control, we questioned whether these were true expression or artifact. To isolate EC and MuSC populations, we first crossed the *Flk1⁺/GFP* mice to label the ECs of the vasculature (*Ema et al., 2006*) and the *Pax7⁺/CreERT2:ROSA26⁺/Loxp-stop-Loxp-tdTomato* (*Pax7⁺/CreERT2:R26R⁺/tdT*) mice to mark the MuSC lineage (*Murphy et al., 2011*; *Verma et al., 2018*). We performed bulk RNA sequencing (RNA-seq) on FACS sorted ECs and MuSCs as well as freshly isolated single muscle fibers (*Figure 1A*, *Figure 1—figure supplement 1E–G*). We found that single muscle fibers routinely have ECs fragments attached to the fiber (*Figure 1—figure supplement 1G*) and so we removed such fibers based on *Flk1GFP* expression from the samples collected for sequencing. We surveyed for canonical genes for each cell type (*Figure 1B*) and found minimal but reliable expression of canonical ECs genes such as *Pecam1*, *Cdh5*, *Kdr*, and *Flt1* in MuSCs.

It is possible that these EC signatures detected in MuSCs were due to small amounts of contaminating ECs with very high expression of the canonical EC genes skewing the average expression in MuSC RNA samples. To rule out this possibility, we performed single-cell RNA-seq (scRNA-seq) on MuSCs and ECs isolated from mouse hind limb muscle from both basal condition and 3 days post intramuscular cardiotoxin (CTX) injury to look at both quiescent and activated MuSCs from the reporter mice specified above (*Figure 1A*). We could reliably delineate injured and activated MuSCs via side and forward scatter (*Figure 1—figure supplement 1H and I*). We FACS isolated cells from both days separately and spiked in 20% of the ECs into the MuSCs from their respective time points, and performed scRNA-seq for each time point (*Figure 1A*). We performed sequencing with ~300 K read/cell compared with the commonly used sequencing with 60 K reads/cell, in order to maximize the possibility of detecting low-abundance transcripts (*Zhang et al., 2020*). In the aggregated dataset, the MuSCs showed low overlap between D0 and D3 owing to the different stages of the myogenic differentiation cycle, while the ECs clusters showed near perfect overlap (*Figure 1C and D*). While drastic morphological changes in ECs have been shown during muscle regeneration (*Hardy et al., 2016*), transcriptomic changes are much more tapered, especially compared with MuSCs (*Latroche et al., 2017*). We were able to deconvolve the quiescent MuSCs from the activated and differentiating MuSCs, ECs, and other cell types from gene signatures. (*Figure 1—figure supplement 1J*). Importantly, data from scRNA-seq were able to recapitulate the minimal expression of canonical EC genes in the MuSC clusters such as *Cdh5* (*Figure 1E*) as seen in our Bulk RNA-seq results (*Figure 1B*). These included the Vegf receptors *Flk1* (*Kdr*) and *Flt1* (*Figure 1E*).

As a quality control measure, we introduced an artificial chromosome into our reference genome with sequences for the three transgene genes; *eGFP* from *Flk1⁺/GFP*, *tdTomato* and *CreERT2* from *Pax7⁺/CreERT2:R26R⁺/tdT* and used this genome to map our single cell RNA-seq data (*Figure 1E*, *Figure 1—figure supplement 1K*). Surprisingly, we also found *eGFP* in the MuSCs and *tdTomato* in EC fraction, while the *CreERT2* expression remained mainly restricted to the MuSCs (*Figure 1—figure supplement 1K*). FACS analysis and FACS-sorted cells confirmed that GFP(+) and tdTomato(+) cells are exclusively restricted as ECs and MuSCs, respectively (*Figure 1—figure supplement 1E and F*). Therefore, we hypothesized that this was due to the ambient-free mRNA from the digested cells that is intrinsic to any droplet based single-cell sequencing platform. By using SoupX (*Young and Behjati, 2020*), we performed careful background subtraction using genes expressed exclusively in myofibers as our negative control and genes validated by in situ hybridization as a positive control (*Kann and Krauss, 2019*). We observed decreased but sustained *eGFP* expression in the MuSC fraction and *tdTomato* expression in the EC fraction after SoupX subtraction (*Figure 1—figure supplement 1K*). In addition, the EC signatures such as *Cdh5* expression in the MuSC fraction was also sustained. These results conclude that MuSCs contain mRNAs from canonical ECs genes. We showed that the canonical EC genes, such as *Cdh5*, *Flt1* and *Flk1* were broadly expressed in the myogenic cells in our dataset (*Figure 1E*).

Since detection of rare subpopulation in single cell dataset is a factor of cell numbers, we re-analyzed previously published dataset with 2,232 myogenic cells across different states (*Torre et al., 2018*; *De Micheli et al., 2020*). We were able to classify cell as quiescent, proliferative vs. differentiating states based on the expression of *Calcr*, *Cdk1*, and *Myog*, respectively (*Figure 1—figure*

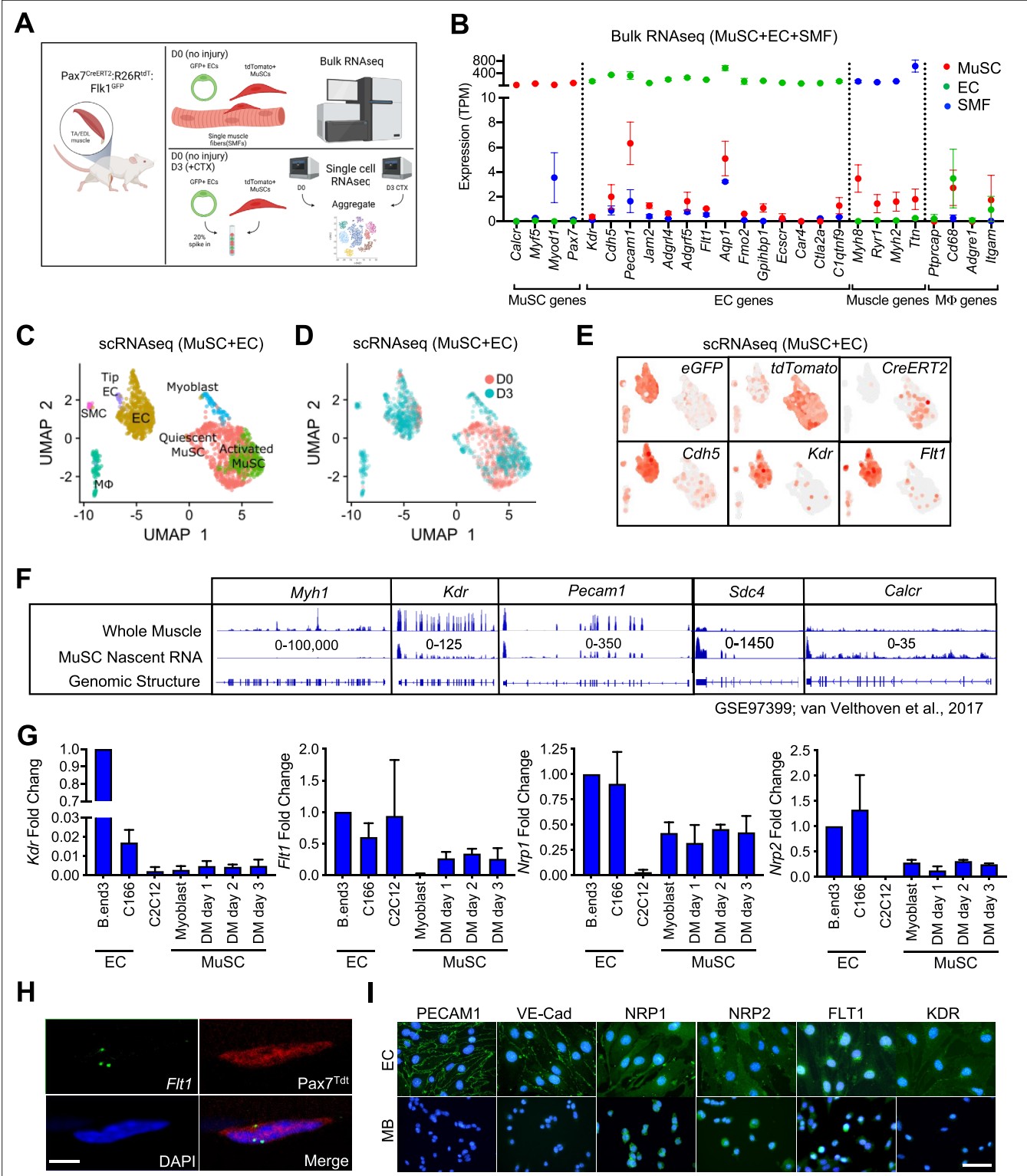

**Figure 1.** EC gene signal including VEGF receptor genes in MuSCs. (**A**) Experimental schema for bulk and scRNA-seq from the *Pax7^CreERT2^:R26R^tdT^:Flk1^GFP^* mice. Bulk RNA-seq performed on MuSCs, ECs and single muscle fibers (SMFs) from uninjured muscle. FACS sorted MuSCs and ECs from uninjured and regenerating TA muscle (3 days following CTX) were run separately on the 10 X single cell platform and aggregated. This panel created with BioRender. com, and published using a BW26O8AXHL license with permission. (**B**) Bulk RNA-seq showing EC signature in MuSCs. Subset dividing genes that are commonly used to delineate cell identity for MuSCs, ECs and SMFs. Last column shows genes that define macrophages (Mφ), which should not be highly expressed in any on our cell types. Red dots indicate MuSCs, green dots indicate ECs and blue dots indicate SMFs. Data show mean ± SD (n=3).

*Figure 1 continued on next page*

Figure 1 continued

(**C**) UMAP from aggregated single cell RNA-seq shows expression of different phases of MuSCs (quiescent MuSCs, activated MuSCs and myoblasts), ECs (tip ECs and ECs) and from likely contaminant cells such as macrophages (Mφ) and smooth muscle cells (SMC). (**D**) UMAP from aggregated data visualized by sample day showing MuSCs segregated by the sample day but overlap in the EC population. Red dots indicate intact (day 0) and blue dots indicate 3 days following CTX. (**E**) Expression of quality control genes such as *eGFP*, *tdTomato*, *CreERT2,* and EC genes such as *Cdh5, Kdr,* and *Flt1*. (**F**) Genome browser tracks of whole muscle and TU-tagged MuSC nascent RNA (GSE97399, *van Velthoven et al., 2017*). *Kdr* and *Pecam1* expression can be found in the MuSC fraction. As control, *Myh1* is highly expressed in the whole muscle preparation but largely absent in the MuSC fraction. *Sdc4* and *Calcr* are highly expressed in MuSC and less so in the whole muscle fraction. (**G**) qPCR for *Kdr, Flt1, Nrp1* and *Nrp2* in EC lines (bEnd.3 and C166), muscle cell line (C2C12) and MuSC-derived myoblasts in growth and differentiation medium (DM) shows low level expression of VEGFRs and VEGF co-receptors. Data show mean ± SD (n=3). (**H**) RNAScope of *Flt1* on freshly isolated single muscle fibers from *Pax7^{tdT}* mice shows *Flt1* expression (green) and tdTomato (red) in MuSCs. Nuclei were counterstained with DAPI (blue). Scale bar indicates 5 µm. (**I**) Immunostaining for PECAM1, VE-cadherin (VE-Cad), VEGFA co-receptors (NRP1 and NRP2) and VEGFA receptors (FLT1 and FLK1) in bEnd.3 EC cell line and MuSC-derived myoblasts (MB). Nuclei were counterstained with DAPI (blue). Scale bar indicates 20 µm.

The online version of this article includes the following source data and figure supplement(s) for figure 1:

**Source data 1.** Measurement of EC gene signal including VEGF receptor genes in MuSCs.

**Figure supplement 1.** EC gene signal in MuSCs.

**Figure supplement 1—source data 1.** Measurement of EC gene signal in MuSCs.

*supplement 1L*). We noticed that EC prototypic markers such as *Flt1* are broadly expressed with small amounts in MuSCs. Complementary data from different laboratories showed the clear expression of EC prototypic markers such as *Cdh5, Flt1,* and *Kdr,* using microarrays and Bulk-RNA-seq (*Figure 1— figure supplement 1A and B*; *Fukada et al., 2007*; *Ryall et al., 2015*). RNA-seq data from fixed quiescent, early activated and late activated MuSCs show that *Flt1* may be transiently upregulated during the early activation process (*Figure 1—figure supplement 1C*; *Yue et al., 2020*). To confirm whether the EC gene mRNAs were transcribed from MuSCs, we utilized previously published MuSC nascent RNA transcriptome from TU-tagged samples (*Gay et al., 2013*; *van Velthoven et al., 2017*). As expected, *Myh1* was represented in the whole muscle but was absent in the TU-tagged MuSCs (*Figure 1F*), indicating that the nascent MuSCs were devoid of cellular contamination from other cells in the muscle. Inversely, the nascent MuSC transcript was over-represented for MuSC related genes such as *Calcr* and *Sdc4*. Interestingly, we were able to detect EC genes such as *Kdr* and *Pecam1* in the TU-tagged MuSC samples indicating that they were actively transcribed by MuSCs (*Figure 1F*, *Figure 1—figure supplement 1D*).

We also verified the expression of *Vegfr* genes (*Kdr*, *Flt1*, *Nrp1*, and *Nrp2*) in MuSCs using RT-qPCR (*Figure 1G*). In addition, we verified the expression of *Flt1* by performing in situ hybridization using RNAScope on MuSC on whole muscle fiber, which we currently believe to be the gold standard for expression studies (*Figure 1H*). Finally, in MuSC-derived myoblasts, NRP1and NRP2 expression was detectable with comparable intensity compared with EC cell line, while FLT1 expression was detectable with lower intensity compared with EC cell line (*Figure 1I*). By contrast, PECAM1, VE-Cadherin and FLK1 expression, which was clearly detected in EC cell line, was undetectable level in myoblasts. Taken together, these data indicate that there are both transcripts of these EC canonical genes and EC canonical proteins in MuSCs.

## VEGFA induces proliferation and cell survival but not differentiation in myoblasts

Since VEGFRs were expressed in MuSCs in small amounts and their ligand, VEGFA, was highly expressed in MuSCs (*Verma et al., 2018*), we wanted to investigate whether there were any biological effects to induction by VEGFA. We found that treatment with VEGFA could increase proliferation of MuSC-derived myoblasts at low dose but inhibit proliferation at high dose of VEGFA, a phenomenon that has been previously described in ECs (*Noren et al., 2016*; *Figure 2—figure supplement 1A*). We saw no effect on differentiation by VEGFA as evaluated by myosin heavy chain (MyHC) staining, fusion index and RT-qPCR (*Figure 2—figure supplement 1B–D*). By contrast, crystal violet staining showed that VEGFA could significantly increase survival as judged by number of myoblasts following UV-mediated apoptotic cell death induction (*Figure 2—figure supplement 1E and F*). To investigate apoptosis

in detail, we optimized Annexin V assay following thapsigargin-mediated endoplasmic reticulum (ER)-stress (*Hirai et al., 2010*) so that we could study deviation at ~ED50 while still performing experiments to remove the confounding variable to proliferation from the experimental setup (*Figure 2—figure supplement 1G and H*). We had previously shown that MuSCs are the predominant cells that secrete VEGFA in skeletal muscle (*Verma et al., 2018*) and while adding exogenous VEGFA did not improve cell survival, blocking VEGFA via a soluble form of FLT1-FC increased the number of apoptotic and necrotic myoblasts in vitro (*Figure 2A and B*).

## VEGFA-facilitated cell survival in MuSC-derived myoblasts is mediated through FLT1

To characterize the VEGF receptor responsible for the anti-apoptotic effect of VEGFA on MuSC-derived myoblasts, we used pharmacological inhibitors of the VEGF receptors (*Figure 2C and D*). We used blocking antibody for the VEGF receptors FLT1 (anti-FLT1 antibody), small molecule inhibitors for FLK1 (SU4502 and ZM306416) and the FLK1 co-receptor NRP1 (EG00229) following thapsigargin induction (*Figure 2A and D*). Surprisingly, inhibiting FLK1, the major signaling RTK for VEGFA, had no effect on myoblasts survival following thapsigargin induction (*Figure 2D*). By contrast, blocking FLT1 via blocking antibody greatly decreased the survival of myoblasts following thapsigargin induction (*Figure 2D*). To confirm this interesting result using genetic tools, we obtained myoblasts with *Pax7-CreER*-inducible deletion of *Flt1* mice (*Pax7$^{+/CreER}$:Flt1$^{Loxp/Loxp}$* or MuSC-*Flt1$^{\Delta/\Delta}$*) and the control mice (*Pax7$^{+/+}$:Flt1$^{Loxp/Loxp}$*). In vitro 4-OHT-mediated genetic deletion of *Flt1* (MuSC-*Flt1$^{\Delta/\Delta}$*) resulted in down-regulation of *Flt1* RNA and FLT1 protein expression (*Figure 2—figure supplement 1I and J*), and increased spontaneous apoptotic cell death even without induction of apoptosis (*Figure 2E*). By contrast, *Flt1* deletion did not affect cell proliferation assessed by EdU staining or myogenic differentiation assessed by MyHC staining (*Figure 2—figure supplement 1K–M*). When thapsigargin-induced apoptosis was induced, the MuSC-*Flt1$^{\Delta/\Delta}$* myoblasts had increased apoptosis that was not responsive to exogenous VEGFA (*Figure 2F*).

## AKT1 signaling is involved in apoptosis of muscle stem cells

VEGFA signaling is mediated through Extracellular signal-Regulated Kinase (ERK), p38 Mitogen-Activated Protein Kinase (MAPK), and Protein kinase B (AKT1). In ECs, VEGFA is known to protect cells from apoptosis via AKT1 (*Domigan et al., 2015*; *Lee et al., 2007*). However, it is not known whether VEGFA can similarly activate AKT1 in MuSC-derived myoblasts. While the role of AKT1 has been explored in proliferation and differentiation in myoblasts, its role in apoptosis has not been well characterized (*Loiben et al., 2017*). We assessed for AKT1 activation via phosphorylated AKT1 (pAKT1) in MuSC-derived myoblasts. We found that exogenous VEGFA could induce AKT1 phosphorylation (pAKT1) (*Figure 2G and H*, *Figure 2—figure supplement 1N and O*). This response was blunted in MuSC-*Flt1$^{\Delta/\Delta}$* myoblasts and was no longer responsive to VEGFA (*Figure 2G and H*, *Figure 2—figure supplement 1N and O*). Lastly, we wanted to confirm that AKT1 activation could improve myoblast survival. We infected lentiviral *E4ORF1* or *MyrAKT1* vectors in myoblasts, both of which gene products have been shown to specifically activate AKT1 without activating ERK or p38 (*Kobayashi et al., 2010*). We found that overexpression of either of these genes improved cell survival compared with the control in vitro following induction of apoptosis via thapsigargin (*Figure 2I*). These data establish FLT1-AKT1 as the cascade linking VEGFA to apoptosis in MuSC-derived myoblasts during muscle regeneration (*Figure 2J*).

## VEGFA-FLT1 pathway protects MuSCs from apoptosis in vivo

Endogenous and exogenous VEGFA have been shown to regulate cell survival and protect ECs from apoptosis (*Gerber et al., 1998*; *Lee et al., 2007*). To assess whether additional VEGFA had an effect on MuSC behaviors in vivo, we used mice carrying the *Vegfa$^{+/Hyper}$* allele for injury-mediated TA muscle regeneration following BaCl$_2$ injection (*Figure 3A and B*; *Miquerol et al., 1999*). MuSC-derived myoblasts from *Pax7$^{+/tdT}$:Vegfa$^{+/Hyper}$* mice showed around 2.8-fold increased expression of *Vegfa* but not the *Vegfr* genes compared with myoblasts from wild-type mice (*Figure 3—figure supplement 1A*). Interestingly, while treatment with VEGFA alone had no effect on apoptosis in vitro, the MuSCs from *Pax7$^{+/tdT}$:Vegfa$^{+/Hyper}$* mice showed decreased cell death in regenerating muscle by 1 day following BaCl$_2$ injection (*Figure 3C*). Consequently, single muscle fibers from *Pax7$^{+/tdT}$:Vegfa$^{+/Hyper}$*

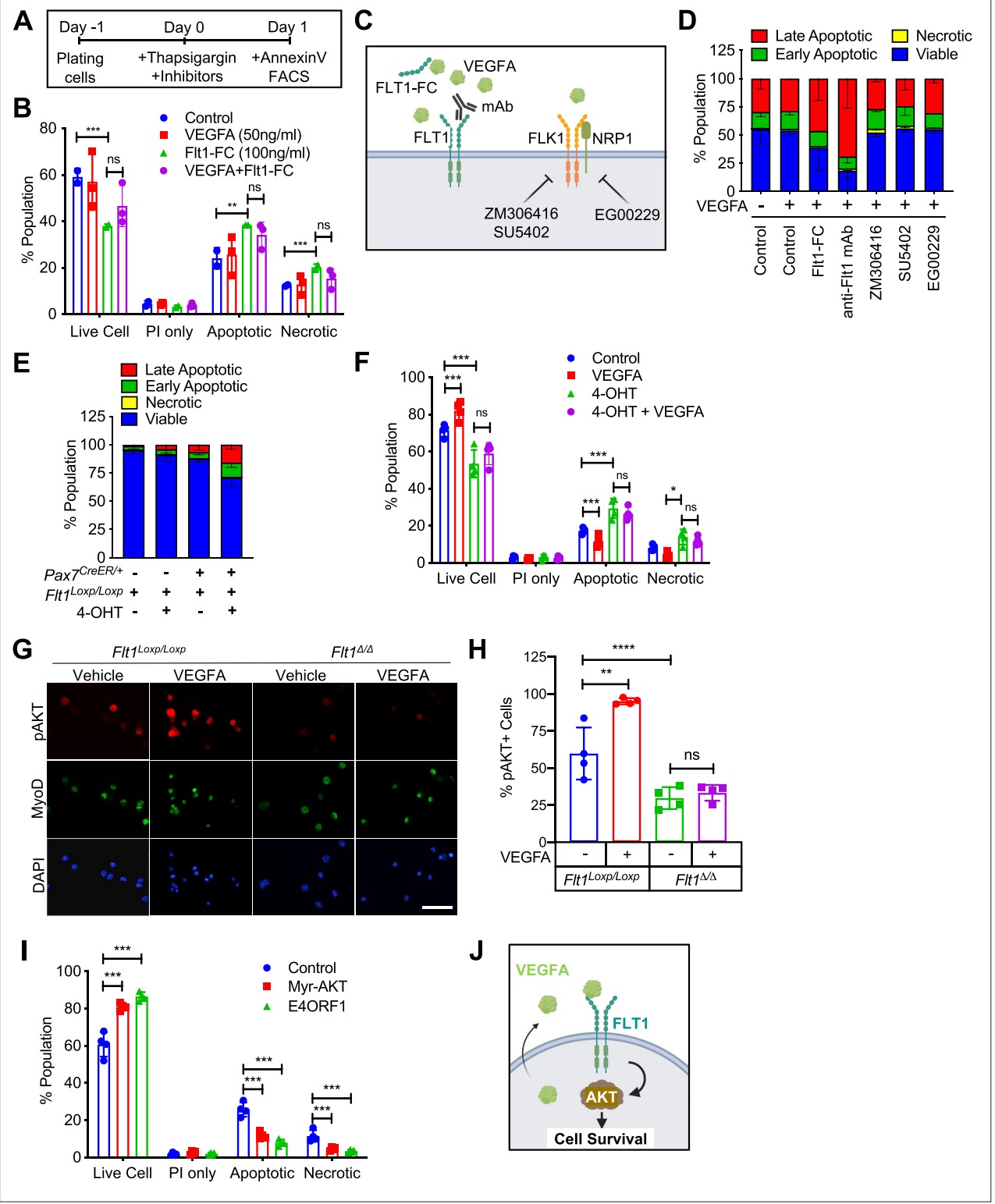

**Figure 2.** VEGFA-FLT1-AKT1 axis controls apoptosis in MuSC in vitro. (**A**) Experimental scheme for assessing apoptosis following thapsigargin induction in myoblast culture. (**B**) Decreased cell survival in myoblast in vitro as VEGFA is blocked using 100 ng/ml FLT1-FC (a VEFGA trap) following thapsigargin induction. This phenotype is partially rescued with exogenous VEGFA (50 ng/ml). Data show mean ± SD (n=3). (**C**) Graphical representation of the various tools used to interrogate the VEGFA pathway in this figure. This panel created with BioRender.com, and published using a TQ26O8B2M7 license with permission. (**D**) Following thapsigargin induction, apoptotic and necrotic cells are increased with inhibition of FLT1 via FLT1-FC or anti-FLT1 antibody (anti-FLT1 mAb) but not FLK1 (SU5402 and ZM306416) or NRP1-FLK1 inhibition (EG00229) following exogenous VEGFA (50 ng/ml). Data show

*Figure 2 continued on next page*

*Figure 2 continued*

mean ± SD (n=3). (**E**) 4-OHT induced deletion of *Flt1* in *Pax7^{+/CreER}:Flt1^{Loxp/Loxp}* myoblasts is sufficient to reduce cell survival in myoblast without induction of apoptosis. Data show mean ± SD (n=3). (**F**) Cell survival is decreased in vitro in myoblast with thapsigargin induction following 4-OHT mediated deletion of *Flt1* in *Pax7^{+/CreER}:Flt1^{Loxp/Loxp}* myoblast that is not rescued by exogenous VEGFA. Blue indicates MuSC-*Flt1^{+/+}*, red indicates MuSC-*Flt1^{+/+}* with 50 ng/ml VEGFA, green indicates MuSC-*Flt1^{Δ/Δ}* and purple indicates MuSC-*Flt1^{Δ/Δ}* with 50 ng/ml VEGFA. Data show mean ± SD (n=3). (**G**) Representative images of pAKT1 (red) in myoblast stained by MyoD (green) in MuSC-*Flt1^{+/+}* and MuSC-*Flt1^{Δ/Δ}* myoblasts induced with exogenous VEGFA. Nuclei were counterstained with DAPI (blue). Scale bar indicates 50 μm. (**H**) Quantification of pAKT1 in myoblasts stained by MyoD in MuSC-*Flt1^{+/+}* and MuSC-*Flt1^{Δ/Δ}* myoblast induced w/wo exogenous VEGFA. VEGFA induction increases pAKT1 in MuSC-*Flt1^{+/+}* myoblasts but this response is lost in MuSC-*Flt1^{Δ/Δ}* myoblasts. Data show mean ± SD (n=3). (**I**) Annexin V quantification of myoblasts transfected with myr-AKT1 and E4ORF1 to activate AKT1 showed increased cell survival of myoblasts following thapsigargin induction. Data show mean ± SD (n=3). (**J**) Representative model for VEGFA-FLT1-AKT1 axis-mediated MuSC survival. This panel created with BioRender.com, and published using a TQ26O8B2M7 license with permission.

The online version of this article includes the following source data and figure supplement(s) for figure 2:

**Source data 1.** Measurement of VEGFA-FLT1-AKT1 axis for apoptosis in MuSC in vitro.

**Figure supplement 1.** VEGFA-FLT1-AKT1 axis for cell survival in MuSC in vitro.

**Figure supplement 1—source data 1.** Measurement of VEGFA-FLT1-AKT1 axis for cell survival in MuSC in vitro.

**Figure supplement 1—source data 2.** Uncropped blotting image of *Figure 2—figure supplement 1N*.

mice showed increased number of MuSCs, compared with those from *Pax7^{+/tdT}:Vegfa^{+/+}* mice by 28 days following BaCl$_2$ injection (**Figure 3D**). In addition, muscle regeneration was promoted in *Vegfa^{+/Hyper}* mice in the early (14 days) and late (28 days) muscle repair processes as judged by fiber diameter and increase in eMHC(+) regenerating muscle fibers (**Figure 3E and F**, **Figure 3—figure supplement 1B–F**, ).

We then performed the reciprocal experiment to investigate the consequence of *Vegfa* loss in MuSCs in vivo, and utilized MuSC-specific *Vegfa* knockout mice (*Pax7^{+/CreER}:Vegfa^{Loxp/Loxp}*). We have previously shown that vasculature in the MuSC-*Vegfa^{Δ/Δ}* mouse muscle is perturbed and the proximity between the MuSC and EC is increased (**Verma et al., 2018**). However, the functional consequences of this remained unknown. We confirmed that clear downregulation of VEGFA protein in MuSC-derived myoblasts isolated from MuSC-*Vegfa^{Δ/Δ}* mice (**Figure 3—figure supplement 2A**). We noticed that deletion of *Vegfa* in MuSCs in the MuSC-*Vegfa^{Δ/Δ}* mouse muscle led to an increase in the proportion of dead MuSCs following BaCl$_2$ injection (**Figure 3G**). Consequently, the number of MuSCs in the MuSC-*Vegfa^{Δ/Δ}* muscle were significantly reduced following recovery after injury without difference in the MuSC numbers in MuSC-*Vegfa^{Δ/Δ}* muscle at homeostasis (**Figure 3H**). In addition, the muscle had a regenerative defect as indicated by the shift in fiber size distribution, decrease in the size of regenerating eMHC(+) fiber and increased adipose following muscle injury (**Figure 3B, I and J**, **Figure 3—figure supplement 2B–E**, **Figure 3—figure supplement 1E and F**). While a limitation of this experiment is that the MuSC fusion into the fiber also deletes *Vegfa* from the fiber themselves, muscle-fiber-specific deletion of *Vegfa* has not shown an effect on fiber size (**Delavar et al., 2014**). These data indicate that cell intrinsic VEGFA improves cell survival of MuSCs and that loss of MuSC-derived VEGFA results in reduced muscle regeneration.

Since FLT1 but not FLK1 was detected in MuSCs and MuSC-derived myoblasts, we asked whether the *Flt1* had an effect on MuSC survival in vivo, we evaluated cell death in MuSCs from MuSC-*Flt1^{Δ/Δ}* mouse muscle. We induced muscle regeneration using BaCl$_2$ for 1 day and assessed for cell death in MuSCs. As seen in vitro, we found that loss of *Flt1* in MuSCs (**Figure 2—figure supplement 1I and J**) resulted in increased cell death during early regeneration (**Figure 3K**). Consequently, single muscle fibers from MuSC-*Flt1^{Δ/Δ}* mice showed a decreased number of MuSCs, compared with those from MuSC-*Flt1^{+/+}* mice by 28 days following BaCl$_2$ injection (**Figure 3L**). We also examined the long-term in vivo consequence of deleting *Flt1* from MuSC. There was no significant muscle phenotype in MuSC-*Flt1^{Δ/Δ}* muscle at homeostasis (**Figure 3B, M and N**). However, following injury, the MuSC-*Flt1^{Δ/Δ}* muscle had a modest regenerative defect as indicated by the shift in fiber size distribution following muscle injury and decrease in size of eMHC(+) regenerating fibers (**Figure 3B, M and N**, **Figure 3—figure supplement 2F–J**, **Figure 3—figure supplement 1E and F**).

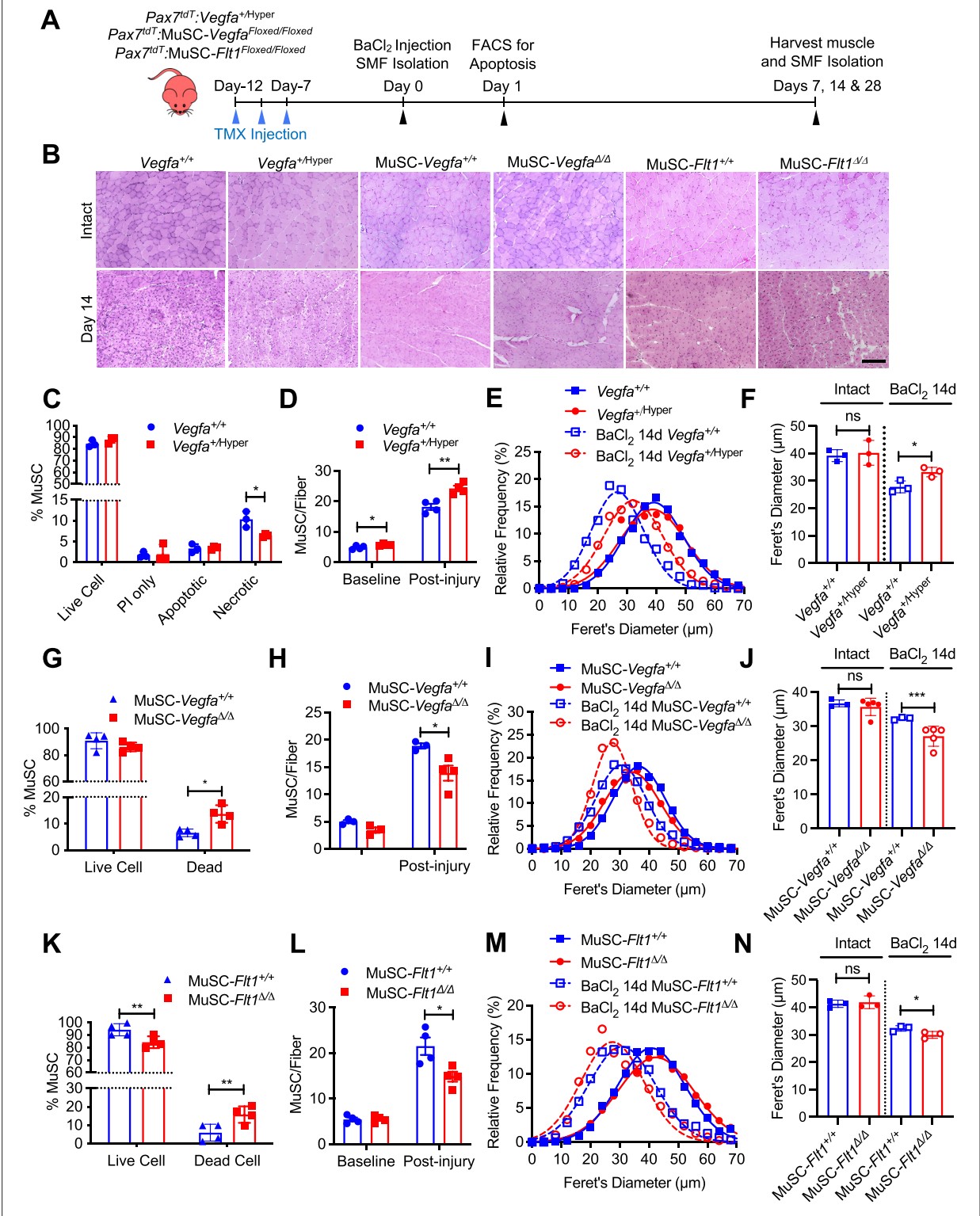

**Figure 3.** MuSC-derived *VEGFA* and *Flt1* requires proper skeletal muscle regeneration. (**A**) Experimental schema detailing the experiments performed in this figure. The *Pax7+/CreER:R26RtdT:Vegfa+/Hyper* (*Vegfa+/Hyper*) *Pax7+/CreER:R26RtdT:VegfaLoxp/Loxp for* MuSC-*VegfaΔ/Δ* and *Pax7tdT:Flt1Loxp/Loxp* for MuSC-*Flt1Δ/Δ* lines were pulsed with tamoxifen (TMX) prior to BaCl₂-induced muscle injury followed by investigations. (**B**) Representative H&E-stained images for intact and on 14-day post injury TA muscle from MuSC-*Vegfa+/Hyper*, MuSC-*Flt1Δ/Δ and* MuSC-*VegfaΔ/Δ* mice and their representative controls. Scale bar indicates 100 μm. (**C**) Annexin V staining show less necrotic cells in MuSC from *Vegfa+/Hyper* mice compared with the control one day following injury. Data show mean ± SD (n=3). (**D**) Quantification of MuSCs from single muscle fibers show increased Pax7 immunofluorescence positive MuSCs in *Vegfa+/Hyper* EDL

*Figure 3 continued on next page*

*Figure 3 continued*

muscle compared with the control mice at base line and 14 days post injury. Data show mean ± SD (n=4). (**E**) Fiber size distribution and (**F**) mean feret's diameter of uninjured and regenerating muscle 14 days post injury from *Vegfa*$^{+/Hyper}$ and control mice show no difference at baseline but an increase in fiber diameter following injury. Data show mean ± SD (n=3). (**G**) Annexin V staining show increased dead cells in MuSCs from MuSC-*Vegfa*$^{Δ/Δ}$ mice one day following BaCl$_2$ compared with the control MuSC-*Vegfa*$^{+/+}$ mice. Data show mean ± SD (n=4). (**H**) Quantification of MuSCs from single muscle fiber at base line and 14 days post injury shows no difference at baseline and reduced MuSC numbers at 14 days. Data show mean ± SD (n=4). (**I**) Fiber size distribution and (**J**) mean feret's diameter of uninjured and regenerating muscle 14 days post injury from MuSC-*Vegfa*$^{Δ/Δ}$ and MuSC-*Vegfa*$^{+/+}$ mice show no difference at baseline but a decrease in fiber diameter following injury. Data show mean ± SD (n=3 or 5). (**K**) Annexin V staining show increased apoptosis in MuSCs from MuSC-*Flt1*$^{Δ/Δ}$ mice one day following injury compared with the control MuSC-*Flt1*$^{+/+}$ mice. Data show mean ± SD (n=4). (**L**) Quantification of MuSCs from single muscle fiber show decreased MuSCs in MuSC-*Flt1*$^{Δ/Δ}$ EDL muscle at base line and 14 days post injury compared with the control MuSC-*Flt1*$^{+/+}$ mice. Data show mean ± SD (n=4). (**M**) Fiber size distribution and (**N**) mean feret's diameter of uninjured and regenerating muscle from MuSC-*Flt1*$^{Δ/Δ}$ and compared with the control MuSC-*Flt1*$^{+/+}$ mice show no difference at baseline but a decrease in fiber diameter following injury. Data show mean ± SD (n=3).

The online version of this article includes the following source data and figure supplement(s) for figure 3:

**Source data 1.** Measurement of MuSC-derived VEGFA and Flt1 for proper skeletal muscle regeneration.

**Figure supplement 1.** MuSC-derived VEGFA and Flt1 for skeletal muscle regeneration.

**Figure supplement 1—source data 1.** Mesurement of MuSC-derived VEGFA and Flt1 for skeletal muscle regeneration.

**Figure supplement 2.** MuSC-derived VEGFA and Flt1 regulating skeletal muscle regeneration.

**Figure supplement 2—source data 1.** Measurement of MuSC-derived VEGFA and Flt1 during skeletal muscle regeneration.

## VEGFA-FLT1 pathway regulates muscle pathology in DMD model mice

While angiogenic defects have been reported in the *mdx* mice as well as in golden retrieval muscular dystrophy (GRMD; canine model of DMD) (*Verma et al., 2010*; *Latroche et al., 2015*; *Verma et al., 2019*; *Kodippili et al., 2021*; *Podkalicka et al., 2021*), it is not clear whether VEGF family and its receptors are implicated in human dystrophinopathies. We probed the VEGF ligands and receptors in microarrays (*Supplementary file 1*) from skeletal muscles and MuSCs from *mdx* mice (*Tseng et al., 2002*; *Pallafacchina et al., 2010*) and skeletal muscles from the GRMD (*Vieira et al., 2015*). *Vegfa* was downregulated in both models (*Figure 4—figure supplement 1A*). *Flt1* was also downregulated in GRMD but not *mdx* muscles. To examine whether VEGF signaling is altered in DMD patients, we performed gene expression analysis on previously available data from microarrays from patients with DMD (*Chen et al., 2000*). We also aggregated and probed microarray data from muscle biopsies of patients with various neuromuscular diseases or of healthy individuals after exercise (*Bakay et al., 2006*). In the microarray data, *Vegfa* expression was increased after an acute bout of exercise, and *Vegfa* expression was reduced in ALS muscle, BMD muscle, as well as both early and late phases of DMD muscle (*Figure 4—figure supplement 1A*). These data indicate that *Vegfa* expression is decreased in dystrophinopathy, and thus increasing VEGFA may be a therapeutic target for DMD.

Therefore, we crossed the MuSC-*Flt1*$^{Δ/Δ}$ mice with the chronically regenerating DMD model mice (*mdx*) to generate *mdx*:MuSC-*Flt1*$^{Δ/Δ}$ mice, and analyzed long-term effects of *Flt1* deletion (*Figure 4A*, *Figure 4—figure supplement 1B and C*). Importantly, we found a significant decrease in fiber diameter, increased fibrosis and CD31(+) capillary density (*Figure 4B–D*, *Figure 4—figure supplement 1C and D*) in TA muscle. This was accompanied by a physiological decrease in muscle perfusion as shown by laser Doppler flow at 12 months (*Figure 4E*) as well as a functional decline in muscle strength as judged by grip strength both acutely and chronically (*Figure 4F*).

By contrast, when we crossed the *Vegfa*$^{+/Hyper}$ mice with *mdx* mice (*Figure 4A*), we noticed a significant increase in fiber diameter, increase in capillary density and decreased fibrosis (*Figure 4G–I*, *Figure 4—figure supplement 1D–F*) in both TA and diaphragm muscle of *mdx*:*Vegfa*$^{+/Hyper}$ mice. This was accompanied by a physiological increase in muscle perfusion as shown by laser Doppler flow at 12 months age (*Figure 4J*) as well as a functional increase in muscle strength as judged by grip strength (*Figure 4K*) without a change in body mass (*Figure 4—figure supplement 1E*). Lastly, when the muscle was injured acutely, with BaCl$_2$, the *Vegfa*$^{+/Hyper}$ mouse had lower number of apoptotic MuSC (*Figure 4—figure supplement 2A–C*). These data indicate that VEGFA-FLT1 axis is a therapeutic target for the pathology seen in the DMD model *mdx* mice.

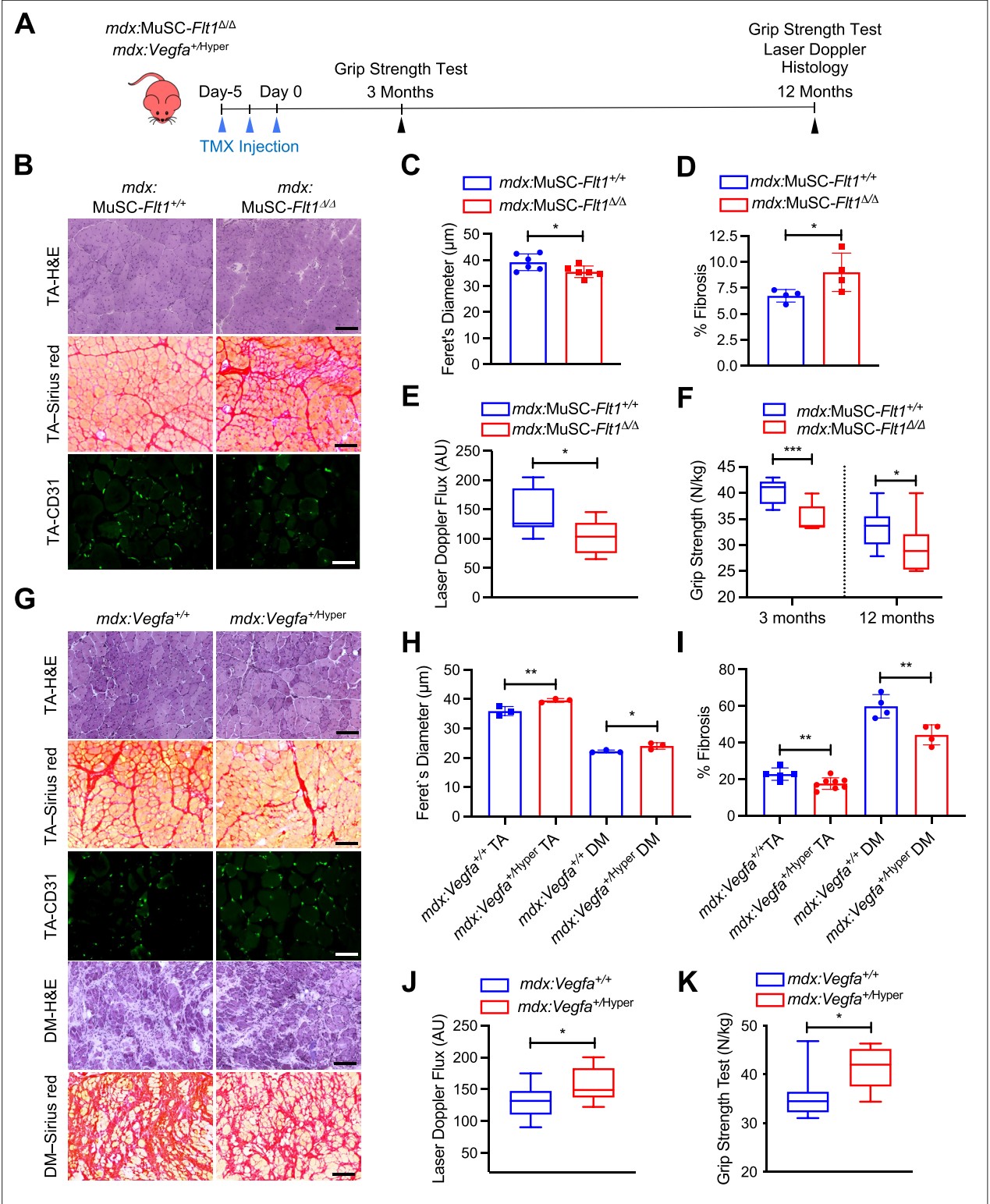

**Figure 4.** VEGFA-FLT1 pathway in MuSCs regulates muscle pathology in DMD model mice. (**A**) Experimental schema detailing the experiments performed in this figure. The *mdx:Pax7^{tdT}:Flt1^{Loxp/Loxp}* was pulsed with tamoxifen (TMX) to generate *mdx*:MuSC-*Flt1^{Δ/Δ}* mice prior to investigation. *mdx:Vegfa^{+/Hyper}* mouse line was used without any induction. (**B**) Representative H&E (scale bar, 100 μm), Sirius red staining (red; scale bar, 100 μm) and CD31(+) capillaries (green; scale bar, 25 μm) from *mdx*:MuSC-*Flt1^{+/+}* and *mdx*:MuSC-*Flt1^{Δ/Δ}* mouse TA muscle at 3 months of age. (**C**) Smaller average fiber size in *mdx*:MuSC-*Flt1^{Δ/Δ}* compared with the control *mdx*:MuSC-*Flt1^{+/+}* mouse TA muscle. Data show mean ± SD (n=6). (**D**) Increased fibrotic area in *mdx*:MuSC-*Flt1^{Δ/Δ}* compared with the control *mdx*:MuSC-*Flt1^{+/+}* mouse TA muscle. Data show mean ± SD (n=4). (**E**) Decreased muscle perfusion

*Figure 4 continued on next page*

*Figure 4 continued*

in *mdx*:MuSC-*Flt1*$^{\Delta/\Delta}$ compared with the control *mdx*:MuSC-*Flt1*$^{+/+}$ mouse TA muscle. Data show mean ± SD (n=3). (n=3). (**F**) Decreased grip strength normalized to body weight in *mdx*:MuSC-*Flt1*$^{\Delta/\Delta}$ compared with the control *mdx*:MuSC-*Flt1*$^{+/+}$ mouse TA muscle at both 3 and 12 months of age. Data show mean ± SD (n=3). (**G**) Representative H&E (scale bars, 100 μm), Sirus red stain staining (red; scale bars, 100 μm) and CD31(+) capillaries (green; scale bar, 25 μm) from TA muscle of *mdx:Vegfa*$^{+/Hyper}$ and *mdx:Vegfa*$^{+/+}$ mouse at 3 months. (**H**) Increased average fiber size in *mdx:Vegfa*$^{+/Hyper}$ compared with the control *mdx:Vegfa*$^{+/+}$ mouse TA and diaphragm (DM) muscle. Data show mean ± SD (n=3). (**I**) Decreased fibrosis in *mdx: Vegfa*$^{+/Hyper}$ compared with the control *mdx:Vegfa*$^{+/+}$ mouse TA muscle and diaphragm (DM) muscle. Data show mean ± SD (n=4 to 8). (**J**) Muscle perfusion is increased in *mdx:Vegfa*$^{+/Hyper}$ compared with the control *mdx:Vegfa*$^{+/+}$ mouse TA muscle. Data show mean ± SD (n=3). (**K**) Grip strength normalized to body weight is increased in *mdx:Vegfa*$^{+/Hyper}$ compared with the control *mdx:Vegfa*$^{+/+}$ mice. Data show mean ± SD (n=3).

The online version of this article includes the following source data and figure supplement(s) for figure 4:

**Source data 1.** Measurement of VEGFA-FLT1 pathway in MuSCs for muscle pathology in DMD model mice.

**Figure supplement 1.** VEGFA-FLT1 pathway in MuSCs for muscle pathology in DMD model mice.

**Figure supplement 1—source data 1.** Measurement of VEGFA-FLT1 pathway in MuSCs for muscle pathology in DMD model mice.

**Figure supplement 2.** VEGFA-FLT1 pathway in MuSCs regulates apoptotic cell death in DMD model mice.

**Figure supplement 2—source data 1.** Measurement of apoptotic cell death.

## Discussion

In this report, we performed bulk and single cell RNA sequencing on MuSCs and ECs. Since deep reads can significantly reduce the effect of the technical noise in scRNA-seq, it can improve estimation of minor transcriptional state of a given cell (*Zhang et al., 2020*). Unexpectedly, we found that MuSCs broadly express EC prototypic markers in small amounts and used multiple different bioinformatics techniques to validate the results. While similar phenomenon in myogenic cells during development and existence of blood-vessel-associated myoendothelial cells in the adult skeletal muscle have been previously described, no functional follow up as been performed leading to the questions whether these minor expression profiles were artifacts or functional (*De Angelis et al., 1999*; *Minasi et al., 2002*; *Tamaki et al., 2002*; *Zheng et al., 2007*; *Roobrouck et al., 2011*; *Huang et al., 2014*; *Charville et al., 2015*; *Goel et al., 2017*; *Giordani et al., 2019*). Our goal was to see whether this small expression pattern had biological consequences. We ultimately decided to use *Flt1* for further investigations and used RNAscope and immunostaining to validate its expression in MuSCs. We found that *Flt1* indeed exerts a biological function even at a low expression. Signaling through VEGFA-FLT1-AKT1 can improve cell survival in MuSCs both in vivo and in vitro.

On a grander scale, our finding of EC prototype markers expressed in MuSC calls into two questions (1) the genes that we used to specify cellular identities and (2) the cellular identity of MuSCs and ECs. The former is important as when we experimentally label, induce or perform Cre-mediated gene knockout experiments based on our assumptions of different gene expression results which may be confounded for these low expressing genes. For example, we have previously investigated both *Flt1* and *Kdr* in mouse muscle using three different reporters and found them to be negative in MuSCs, thereby disregarding their cell-autonomous effect when evaluating global knockouts (*Verma et al., 2010*; *Verma et al., 2018*). It is also possible that EC mRNAs are results of transcription from the cell or a result of mRNA transfer from neighboring cells (*Desrochers et al., 2016*). Of note, the transmission of *tdTomato* mRNA and protein from *Pax7*$^{+/CreERT2}$:*R26R*$^{+/tdT}$ mice used in this study has been recently shown via exosome, opening up the possibility of transmission of other mRNA from MuSC to ECs (*Murach et al., 2020*). The later is an interesting phenomenon form a developmental point of view. MuSCs and ECs arise from a bipotent progenitor originated from somites during early development (*Kardon et al., 2002*; *Hutcheson and Kardon, 2009*; *Lagha et al., 2009*; *Mayeuf-Louchart et al., 2014*; *Mayeuf-Louchart et al., 2016*). Therefore, it is possible that there is a permissive chromatin state that allows for expression of reciprocal genes in the two populations. Along the lines of these observations, FLK1(+) or VE-cadherin(+) cells can contribute to myogenic cells in vitro and after cell transplantation (*Tamaki et al., 2002*; *Le Grand et al., 2004*; *Zheng et al., 2007*; *Huang et al., 2014*), and during development (*Motoike et al., 2003*; *Mayeuf-Louchart et al., 2014*; *Drummond and Hatley, 2018*). Important notion is that the PDGFRα(-)FLK1(+) population exhibited restricted potential to differentiate into the MuSCs in injured muscle (*Sakurai et al., 2008*). Interestingly, in the zebrafish, exogenous expression of *Etv2* in the fast muscle can lead to transdifferentiation of muscle fibers into functional vessels so there is evidence of cell fate flexibility (*Veldman et al., 2013*). The

potential of EC transdifferentiation was also examined by ETV2 overexpression in five human cell types, skeletal muscle cells, adipose-derived mesenchymal stem cells, umbilical-cord-derived mesenchymal stem cells, embryonic lung fibroblast cells, and skin fibroblast cells. Among them, human skeletal muscle cells showed the highest amenability for this EC induction following infection with ETV2 lentivirus vector (*Yan et al., 2019*). Conversely, *Etv2*-deficient vascular progenitors can differentiate into skeletal muscle cells (*Chestnut et al., 2020*). It would be interesting to see whether other EC gene signatures also have functional consequences in the MuSC or muscle at large.

We decided to focus on function of *Flt1* among several EC genes expressed in MuSCs for further investigations on MuSC biology. Our pharmacological and genetic analyses demonstrate that MuSC-derived VEGFA has a drastic effect on cell survival in the via its receptor FLT1 by signaling through AKT1. While VEGFA binds to both FLT1 and FLK1, VEGFB and PGF only bind to FLT1. This creates a scenario where PGF and VEGFB binding can sequester FLT1, increasing free VEGFA availability for VEGFA-FLK1 binding which is the major VEGF signaling pathway for many cell types (*Vempati et al., 2014*). While PGF is not normally expressed in adult tissues, VEGFB is expressed in the MuSCs and muscle fiber (data not shown). Importantly, the VEGFB-FLT1 axis has also been shown to inhibit apoptosis in retina and brain cells in mouse models of ocular neurodegeneration and stroke (*Li et al., 2008*). While our results cannot rule out the involvement of VEGFB in protection of MuSC apoptosis, we provide evidence from both pharmacological and genetic data to indicate that VEGFA is involved.

Despite drastic effect of VEGFA-FLT1 on apoptosis in vitro, the long-term consequences of in vivo deletion of *Flt1* in the MuSC compartment were modest compared with deletion of *Vegfa* in the MuSCs unless crossing with *mdx* mice. Although *Vegfa* is required for both MuSC survival and recruitment of vascular niche (*Verma et al., 2018*), in the steady state, the MuSC turnover may be low enough that the apoptotic stress burden is low. We demonstrated that VEGFA improves cell survival during the proliferative stage following injury in *mdx:Vegfa$^{+/Hyper}$* mice. The evidence that MuSC survival is impaired comes indirectly from transplantation experiments where MuSCs obtained from *mdx* mice engraft to a much lesser degree than those from WT mice (*Boldrin et al., 2015*).

During the review process of this paper, two additional labs reported complementary findings about VEGFA signaling in MuSC (*Chen et al., 2022*; *Groppa et al., 2023*). Groppa et al performed extensive transcriptomics to show role of the VEGFA system in muscle regeneration. They showed that VEGFA is expressed in expression in MuSC and inflammatory cells. In addition, VEGFA increased MuSC proliferation through KDR (FLK1). They also found that deletion of MuSC derived VEGFA lead to an increase in TUNEL(+) apoptotic MuSC. The role of KDR in MuSC was also corroborated by Chen et.al who showed that MuSC lacking KDR showed asymmetrical division deficits and limit MuSC proliferation resulting in impaired tissue regeneration. While both these reports focus on the role of VEGFA-KDR on proliferation, the current paper focuses on the role of VEGFA-FLT1 on MuSC survival.

VEGFA and FLT1 targeted therapies are being explored as both pro- and anti-angiogenic therapies for several indications including retinal degeneration, cancer, pre-eclampsia, and neuromuscular diseases (*Bae et al., 2005*; *Verma et al., 2010*; *Mac Gabhann et al., 2010*; *Keifer et al., 2014*; *Verma et al., 2019*; *Bosco et al., 2021*; *Xin et al., 2021*). As these therapies mature, it will be important to ascertain the MuSC-specific effects of VEGFA and FLT1.

# Materials and methods
## Mice

*Flt1$^{LoxP/LoxP}$* were obtained from Gua-Hua Fong (*Ho et al., 2012*). *B6.Cg-Pax7$^{tm1(cre/ERT2)Gaka/J}$* (*Pax7$^{+/CreERT2}$*; JAX stock# 017763; *Murphy et al., 2011*), *B6.Cg-Gt(ROSA)$^{26Sortm9(CAG-tdTomato)Hze/J}$* (*Ai9*; JAX stock # 007909; *Madisen et al., 2010*), *Vegfa$^{+/Hyper}$* (*Vegfatm1.1Nagy/J*; JAX stock# 027314; *Miquerol et al., 1999*) and *B6Ros.Cg-Dmd$^{mdx-5cv}$/J* (*mdx$^{5cv}$*; JAX stock #002379; *Chapman et al., 1989*) were obtained from Jackson Laboratory. *Kdr$^{tm2.1Jrt/J}$* (*Flk1$^{+/GFP}$*) were obtained from Masatsugu Ema (*Ema et al., 2006*). *B6.Cg-Pax7$^{tm1(cre/ERT2)Gaka/J}$* (*Pax7$^{+/CreERT2}$*) mice were crossed with the *B6.Cg-Gt(ROSA)$^{26Sortm9(CAG-tdTomato)Hze/J}$* (*Ai9*) to yield the *Pax7$^{+/CreERT2}$:R26R$^{tdT}$*(*Pax7$^{tdT}$*) mice. *Pax7$^{tdT}$* mice were bred with the *Vegfa$^{+/Hyper}$* and *Flk1$^{+/GFP}$* to obtain *Pax7$^{tdT}$:Vegfa$^{+/Hyper}$* and *Pax7$^{tdT}$:Flk1$^{+/GFP}$* mice. *Vegfa$^{LoxP/LoxP}$* mice obtained from Napoleone Ferrara (*Gerber et al., 1999*) were crossed with *Pax7$^{+/CreERT2}$* to yield the *Pax7$^{+/CreERT}$:Vegfa$^{LoxP/LoxP}$* mice. *Flt1$^{LoxP/LoxP}$* mice obtained from Guo-Hua Fong (*Gerber et al., 1999*) were crossed with *Pax7$^{+/CreERT2}$* to yield the *Pax7$^{+/CreERT}$:Flt1$^{LoxP/LoxP}$* mice. Colonies for all the mice were established

in the laboratory. Cre recombination was induced using tamoxifen (T5648, MilliporeSigma) dosed as 75 mg/kg body weight x 3 times over 1 week at 3–6 weeks of age. Mice carrying the wild-type *CreERT2* allele were used for control experiments. TA muscle regeneration was induced by intramuscular injection of 20 µl of 1% $BaCl_2$ (342920, MilliporeSigma) or 20 µl of 10 µM Cardiotoxin (CTX) (V9125, MilliporeSigma). Mice used for this study is summarized in Key Resources Table.

Genotyping to detect the transgenic and mutant alleles was performed by PCR using the primers described on the web site of Jackson Laboratory shown in Key Resources Table. All primers were synthesized as custom DNA oligos from Integrated DNA technologies (IDT). Genotyping to detect the mutated allele of *mdx$^{5cv}$* was performed by PCR using the primers (0981 and 0982) shown in Key Resources Table. The PCR product DNA was digested with *Dra*II restriction enzyme (R3510S, New England Biolabs). Wild-type allele generated 180 bp and mutant allele generated 50 and 130 bp bands.

The animals were housed in an SPF environment and were monitored by the Research Animal Resources (RAR) of the University of Minnesota. All protocols (2204–39969A) were approved by the Institutional Animal Care and Usage Committee (IACUC) of the University of Minnesota and complied with the NIH guidelines for the use of animals in research.

## Cell isolation by FACS

*Pax7$^{tdT}$:Flk1$^{GFP}$* mice were utilized for FACS-mediated MuSC and EC isolation as previously described (*Asakura et al., 2002*; *Verma et al., 2018*). We performed extensive validation of the fluorescent reporter mice as previously described (*Figure 1—figure supplement 1A–C*; *Verma et al., 2018*). Briefly, quiescent MuSCs and ECs were isolated from the hind limb skeletal muscle of 1- to 2-month-old *Pax7$^{tdT}$:Flk1$^{GFP}$* mice after digestion with collagenase type II. FACS was performed on an FACS sorter (BD FACSAria) and data were analyzed using FlowJo (BD Biosciences). Sorting gates, tdTomato(+) for MuSCs and GFP(+) for ECs, were strictly defined based on control cells isolated from wild-type mice and the forward scatter and side scatter gating. Sorted cells were immediately characterized by immunostaining on slide glasses, utilized for RNA preparation or cultured on collagen-coated plates in the myoblast growth medium as below to obtain MuSC-derived myoblasts and ECs. FACS analysis was performed as previously described (*Turaç et al., 2013*). Cells were either trypsinized (cultured cells) or a single cell suspension was obtained following enzymatic digestion as whole muscle-derived cells (*Asakura et al., 2001*; *Asakura et al., 2002*). Cells were then washed with FACS buffer (2% BSA and 1 mM EDTA in PBS) followed by live/dead staining using ZombieNIR (423105, Biolegends). Cells were washed, then immunostained for cell surface markers. Blocking cells was performed with 1% BSA/PBS, and cells were incubated in fluorescently-conjugated antibody. FACS was performed on a Fortessa X-20 (BD Biosciences) with a 355 nm, 405 nm, 488 nm, 561 nm, and 640 nm lasers.

## Cell culture

Mouse bEnd.3 EC cells (CRL-229), C166 EC cells (CRL-2581), and C2C12 myoblast cells (CRL-1772) were obtained from American Type Culture Collection (ATCC). Human 293 FT cells (R70007) were obtained from ThermoFisher Scientific. All cell lines were cultured in DMEM medium with 10% FBS, 100 units/ml of penicillin, and 100 µg of streptomycin at 37 °C in 5% $O_2$ and 5% $CO_2$. All cell lines were STR profiled to confirm their identity and tested negative for mycoplasma. MuSC-derived myoblast isolation from adult mice was performed as previously described (*Motohashi et al., 2014*). Briefly, after collagenase type II (CLS-2, Worthington) treatment, dissociated cells from mouse hindlimb muscle were incubated with anti-CD31-PE (12-0311-82, eBiosciences), anti-CD45-PE (12-0451-81, eBiosciences), anti-Sca1-PE (A18486, eBiosciences) and anti-Integrin α7 (ABIN487462, MBL International), followed by anti-PE microbeads (130-048-801, Miltenyi Biotec), and then performed LD column (130-042-901, Miltenyi Biotec) separation. Negative cell populations will be incubated with anti-Mouse IgG beads (130-048-402, Miltenyi Biotec), and then MS column (130-042-201, Miltenyi Biotec) separation was performed to isolate Integrin α7(+) MuSCs. MuSC-derived myoblasts were maintained in culture on collagen coated plates in myoblast medium containing 20% FBS, 20 ng/ml bFGF (PHG0263, Invitrogen), 100 units/ml of penicillin and 100 µg of streptomycin in HAM's-F10 medium. Cell cultures were maintained in a humidified incubator at 37 °C with 5% $CO_2$ and 5% $O_2$. 4-Hydroxy tamoxifen (4-OHT, H6278, MilliporeSigma) treatment (1 µM in EtOH) was used to induce *Flt1* deletion in myoblasts isolated from *Flt1$^{LoxP/LoxP}$:Pax7$^{CreERT2}$* mice. For cell survival assay,

1 x 10$^5$ cells were allowed to adhere for 1 day and starved overnight in 0.1% FBS in HAM's F10 medium. Then, cells were exposed to 1 µM EdU along with or without 2–100 ng/ml recombinant VEGFA (493 MV, R&D Systems) for 8 hr before being fixed and stained by the Click-iT EdU Alexa Fluor 488 Imaging Kit (C10337, Thermo Fisher Scientific). For induction of apoptosis in myoblasts, (1–2 x 10$^5$) cells were allowed to adhere to the plates for 16 hr. Thapsigargin-mediated apoptosis was induced by 1 µM of thapsigargin (T9033, MilliporeSigma) dissolved in EtOH with or without VEGFA, 100 ng/ml recombinant FLT1-FC (7756-FL, R&D Systems), 1 µg/ml anti-FLT1 monoclonal antibody (Angio-Proteomie, MAB7072), inhibitors of FLK1, 3 µM ZM306416 (2499/1, R&D Systems) and 10 µM of SU5402 (3300/1, R&D Systems) and an inhibitor of NRP1, 30 µM of EG00229 (6986/10, R&D Systems) for 24 hours. UV light-mediated apoptosis was induced by exposing the cells to UV light in cell culture hood for 45 s without medium. After UV exposure, cell survival was assessed 24 hr following culture in 0.1% FBS in HAM's F10 medium with or without VEGFA using the Crystal violet Assay Kit (ab232855, Abcam) and quantified the Crystal violet dye after solubilization by absorbance at 570 nm. To induce differentiation of myoblasts, the myoblast medium was replaced with differentiation medium that contained DMEM supplemented with 5% horse serum with or without VEGFA or bFGF for 1 or 3 days followed by anti-sarcomeric myosin heavy chain antibody (MF-20, Developmental Study Hybridoma Bank).

## AKT1 induction

The lentiviral pCCL-E4ORF1 and pCCL-myrAkt1 constructs were a kind gift from Dr. Jason Butler (*Kobayashi et al., 2010*). A total of 293 FT cells (R70007, Thermo Fisher Scientific) were seeded in DMEM with 10% FBS and transfected with the lentivirus vectors along with pCMV-VSV-G (8454, Addgene), pRSV-Rev (12253, Addgene), and pMDLg/pRRE (12251, Addgene) using PolyJet transfection reagent (SL100688, Signagen Laboratories). The culture supernatant of the transfected 293 FT cells was added to MuSC-derived myoblast culture with 0.8 µg/ml polybrene (MilliporeSigma, H9268). pAKT1(+) cells were stained with anti-pAKT1 antibody (4060, Cell Signaling).

## Western blotting

Protein extracts of MuSC-derived myoblast culture obtained with an NE-PER Nuclear and Cytoplasmic Extraction reagents (78833, Thermo Fisher Scientific) was used for western blotting. Protein concentration was determined by the Micro BCA Protein Assay Reagent kit (Thermo Fisher Scientific). Following electrophoresis, the proteins were transferred to an Immobilon P membrane (IPVH00010, EMD Millipore) overnight. pAKT1 was detected by Western blotting with anti-pAKT1 antibody (4060, Cell Signaling) followed by anti–rabbit IgG HRP (31460, Cell Signaling Technology). To verify equal loading proteins, the same blots were stripped and reprobed with anti-GAPDH HRP conjugated (3683, Cell Signaling) as a cytosolic marker. The reaction was developed using SuperSignal West Femto chemiluminescent substrate (PI37074, Fisher Scientific) in accordance with the manufacturer's instructions. Protein signals were detected and quantitated by iBright FL1500 (A44241, ThermoFisher Scientific).

## Apoptosis assay

Apoptosis was measured using measured using Annexin V-Biotin Apoptosis Detection Kit (BMS500BT-100, eBioscience) as per the manufacture's instruction. Streptavidin-conjugated Alexa-Fluro-488 was used for detection. Propidium Iodide (PI) was used in all assays except when Pax7tdT(+) cells were utilized or when ZombieNIR (423105, Biolegends) was used. FACS was performed on a Fortessa X-20 (BD Biosciences) equipped with a 355 nm, 405 nm, 488 nm, 561 nm, and 640 nm lasers. For detection of apoptotic cells in TA muscle sections 3 days following BaCl$_2$ injection, TMX was injected into *mdx:Vegfa$^{+/+}$:Pax7$^{tdT}$* and *mdx:Vegfa$^{+/Hyper}$:Pax7$^{tdT}$* mice before BaCl$_2$ injection to label MuSCs during muscle regeneration. To detect apoptotic cells, the TA muscle sections were incubated with anti–activated caspase-3 antibody (ab214430, Abcam) followed by anti–rabbit Alexa-488 antibodies (A11008, ThermoFisher Scientific) for double immunostaining. DAPI (10236276001, MilliporeSigma) was used for counterstaining of nuclei. Microscopic images were captured by a DP-1 digital camera attached to BX51 fluorescence microscope with 10×, 20×or 40×UPlanFLN objectives with cellSens Entry 1.11 (all from Olympus).

## Immunostaining of cells

Immunostaining for PECAM1, VE-Cadherin, VEGFA, VEGFRs was performed on collagen coated coverslips. Other immunostaining was performed on 35 mm tissue culture plates. Cells were fixed with 2% PFA for 5 min and immunostained as previously described (*Verma et al., 2010*). For membrane receptor staining, cells were permeabilized with 0.01% saponin (ICN10285525, ThermoFisher Scientific) which was kept in the staining solution until the primary antibodies were washed off. At which time, 0.01% Triton-X was added to all the buffers. The antibodies used for this study are listed in Key Resources Table.

## Single muscle fiber isolation and staining

Extensor digitorum longus (EDL) muscle was dissected and digested with 0.2% collagenase type I (C0130, MilliporeSigma) for single muscle fiber isolation as previously described (*Verma et al., 2010*). Single muscle fibers were fixed with 2% PFA/PBS, permeabilized with 0.2% Triton-X100 and counterstained with DAPI. Anti-Pax7 antibody(+) or tdTomato(+) MuSCs per single muscle fiber were counted manually.

## RNAscope

RNAscope for *Flt1* transcripts was performed as previously described (*Kann and Krauss, 2019*) on single muscle fibers from *Pax7^{tdT}* mice using the RNAscope Probe - Mm-Flt1 (C1) (415541, ACDBio). Briefly, isolated EDL fibers are fixed in 4% PFA, washed with PBS, and dehydrated in 100% methanol. Subsequently, fibers are rehydrated in a stepwise gradient of decreasing methanol concentrations in PBS/0.1% Tween-20. Fibers are treated with a proteinase for 10 min, followed by hybridization, amplification, and fluorophore conjugation steps.

## Histology and immunostaining for sections

The mouse tibialis anterior (TA) muscle was used for all histological analysis. Tissues were frozen fresh using LiN$_2$ chilled isopentane and stored at –80 °C. Eight µm thick transverse cryosections were used for all histological analysis. Hematoxylin & Eosin (HE) staining were performed as previously described (*Verma et al., 2010*). Sirius red (Direct Red 80, 365548, MilliporeSigma) staining was performed for muscle sections for fibrosis as previously described (*Shimizu-Motohashi et al., 2015*). Muscle sections were stained in Oil Red O solution (O1391-250ML, MilliporeSigma) as previously described (*Wang et al., 2017*). Anti-eMHC (F1.652, Developmental Study Hybridoma Bank) and anti-Laminin (L0663, MilliporeSigma) antibodies followed by anti-mouse Alexa-488 (A11001, ThermoFisher Scientific) and anti-rat Alexa-568 antibodies (A11077, ThermoFisher Scientific) were used for detection of regenerating muscle fibers. For capillary density measurement, anti-CD31 antibody (550274, BD Biosciences) was used for TA muscle sections followed by anti-rat Alexa-488 (A11006, ThermoFisher Scientific). Microscopic images were captured by a DP-1 digital camera attached to BX51 fluorescence microscope with 10×, 20×or 40×UPlanFLN objectives with cellSens Entry 1.11 (all from Olympus). Photoshop (Adobe) and Fiji (NIH) were used for image processing and manually enumerating the fiber feret's diameter (*Schindelin et al., 2012*).

## Grip strength test

Forelimb grip strength test was performed following a previously published procedure (*Aartsma-Rus and van Putten, 2014*). Briefly, mice were gently pulled by the tail after fore limb-grasping a metal bar attached to a force transducer (Grip Strength Meter, 1027CSM-D52, Columbus Instruments). Grip strength tests were performed by the same blinded examiner. Five consecutive grip strength tests were recorded, and then mice were returned to the cage for a resting period of 20 min. Then, three series of pulls were performed each followed by 20 min resting period. The average of the three highest values out of the 15 values collected was normalized to the body weight for comparison.

## Muscle perfusion

RBC flux was evaluated using the moorLabTM laser Doppler flow meter as previously described (*Verma et al., 2010*) with the MP7a probe that allows for collecting light from a deeper tissue level than standard probes according to the manufacturer's instructions (Moor Instruments). The fur from the right hind leg was removed using a chemical depilatory. Readings were taken using the probe

from at least 10 different spots on the TA muscle. The AU was determined as the average AU value during a plateau phase of each measurement.

## RNA and genomic DNA isolation and qPCR

Cultured cells were washed with ice cold PBS and lysed on the place with Trizol. RNA was isolated using the DirectZol RNA Microprep Kit (R2062, Zymo Research) with on-column DNase digestion followed by cDNA synthesis using the Transcriptor First Strand cDNA synthesis kit (04379012001, Roche Molecular Diagnostics) using random primers. Genomic DNA for genotyping was isolated from mouse tail snips with lysis buffer containing Proteinase K (P2308, MilliporeSigma). qPCR was performed using GoTaq qPCR Master Mix (A6002, Promega). The input RNA amount was normalized across all samples and *18 S rRNA* or *HtatsF1* was used for normalization of qPCR across samples. Primer sequences are listed in Key Resources Table. All primers were synthesized as custom DNA oligos from Integrated DNA technologies (IDT).

## Single-cell RNA sequencing and analysis

Cells for single-cell RNA-seq (scRNA-seq) were obtained from hind limb muscles of 2–3 month-old $Pax7^{tdT}:Flk1^{GFP}$ mice following enzymatic digestion as previously described (*Liu et al., 2015*). Dead cells were excluded from the analysis using ZombieNIR (423105, Biolegends). TdTomato(+) and GFP(+) cells were sorted individually and then 20% of GFP(+) cells were spiked into 80% tdTomato(+). We loaded ~5000 cells into 1 channel of the Chromium system for each of these samples and prepared libraries according to the manufacturer's protocol using version 2.0 chemistry (10 x Genomics). Following capture and lysis, we synthesized cDNA and amplified for 12 cycles as per the manufacturer's protocol (10 X Genomics). The amplified cDNA was used to construct Illumina sequencing libraries that were each sequenced with ~300 K read/cell on one lane of an Illumina HiSeq 2500 machine. We used Cell Ranger 3.1 (10X Genomics) to process raw sequencing data. For A custom genome was constructed to include *eGFP-SV40*, *tdTomato-WPRE-BGHPolyA* and *Pax7-IRES-CreERT2* transgenes. Detailed step-by-step instructions can be found at https://github.com/verma014/10XCustomRef, (copy archived at *Verma, 2020*). We carried out analyses of the filtered data using Seurat suite version 3.0 *Stuart et al., 2019* in *RStudio Team, 2020*. For cell imputation, we utilized ALRA through the Seurat wrapper with default settings (*Linderman et al., 2022*). Additional scRNA-seq datasets were obtained from GEO and analyzed using the same method as listed above. A myogenic score was calculated based on the expression of *Myog*, *Pax7*, *Myod1,* and *Myf5*. Step-by-step instructions for the analysis can be found on https://github.com/verma014/10XCustomRef, (copy archived at *Verma, 2020*).

## Background subtraction

10 x Genomics scRNA-seq platform uses many more droplets than cells and so following a run, there are many droplets that do not have cells. These droplets still get sequenced with some of the RNA that is in the solution. This floating RNA can be used to estimate the 'background' in each droplet. A better description of this can be found by the developers of 'SoupX' (*Young and Behjati, 2020*). Since *Cdh5* expression has previously been verified in MuSCs using RNAscope, we were able to use it as a positive control to remove the background or 'soup' from our data. If *Cdh5* is absent from MuSC, we know that the background subtraction was too aggressive and that subtracting the Soup is not reliable in our case. In addition, we know certain genes that are considered to be specific for MuSCs, muscle ECs or muscle fibers based on the bulk RNA-seq (*Verma et al., 2018*). The top 5 genes that are specific to these population (and also detected by 10 x) were selected and used to show the background in our data set was 14.40% and 13.89% for the D0 and D3 dataset, respectively. The step-by-step instructions can be found on https://github.com/verma014/10XCustomRef, (copy archived at *Verma, 2020*).

## Bulk RNA-seq and microarray analysis

FASTQ files were downloaded from SRA using SRA-toolkit. Sequences were trimmed using trimmomatic to remove adapter contamination and low-quality reads. Trimmed sequences were mapped to mouse mm10 using Hisat2 (*Pertea et al., 2016*). Transcript assembly was performed using StringTie (*Pertea et al., 2016*). Cell type specificity was determined as previously described (*Verma et al.,*

*2018*). Microarray analysis was performed using the Affymetrix Transcriptome Analysis Console (TAC). Samples in each experiment were RNA normalized and the expression was acquired using the Affeymetrix Expression analysis console with gene level expression. Heatmaps were generated in Prism 9 (Graphpad, La Jolla, CA).

## Quantification and statistical analysis

Statistical analysis was performed using Prism 9 (Graphpad, La Jolla, CA) or RStudio (**RStudio Team, 2020**). For comparison between two groups, an unpaired T-test was used. For comparison between multiple groups, a one-way ANOVA was used with multiple comparisons to the control. Distributions were compared using a chi-squared test. Graphing of the data was performed using Prism 9. Vector diagrams were modified using Graphic (Autodesk). All values are means ± SD unless noted otherwise. * indicates $p<0.05$, ** indicates $p<0.01$, *** indicates $p<0.001$.

## Acknowledgements

We thank Minnesota Supercomputing Institute (MSI), University of Minnesota Imaging Center (UIC), University of Minnesota FACS Facility, and University of Minnesota Genomics Center (UMGC) for providing data for this paper. We also thank Jake Trask for critical reading of this paper. We thank Drs. Yosuke Mukouyama (National Institute of Health), Napoleone Ferrara (Genentech), Guo-Hua Fong (University of Connecticut) and Masatsugu Ema (Siga University of Medical Science) for providing *Vegfa*-LoxP/LoxP, *Flt1*LoxP/LoxP and *Flk1-GFP* mice, respectively. This work was supported by NIHT32-GM008244 and NIHF30AR066454 to MV, NIAMS grant AR070231 to RSK, a fellowship of the Training Program in Stem Cell Research from the New York State Department of Health to A.P.K. (NYSTEM-C32561GG) and NIHR01AR062142, NIHR21AR070319, MDA Research Grant, and Regenerative Medicine Minnesota (RMM) Grant to AA.

## Additional information

### Funding

| Funder | Grant reference number | Author |
| --- | --- | --- |
| National Institutes of Health | NIHT32-GM008244 | Mayank Verma |
| National Institutes of Health | NIHF30AR066454 | Mayank Verma |
| National Institute of Arthritis and Musculoskeletal and Skin Diseases | AR070231 | Robert S Krauss |
| New York State Stem Cell Science | NYSTEM-C32561GG | Allison P Kann |
| National Institute of Arthritis and Musculoskeletal and Skin Diseases | NIHR01AR062142 | Atsushi Asakura |
| National Institute of Arthritis and Musculoskeletal and Skin Diseases | NIHR21AR070319 | Atsushi Asakura |
| Muscular Dystrophy Association | MDA241600 | Atsushi Asakura |
| Regenerative Medicine Minnesota | RMM 092319 TR 010 | Atsushi Asakura |

The funders had no role in study design, data collection and interpretation, or the decision to submit the work for publication.

## Author contributions

Mayank Verma, Conceptualization, Data curation, Software, Formal analysis, Validation, Investigation, Visualization, Methodology, Writing - original draft, Writing - review and editing; Yoko Asakura, Xuerui Wang, Kasey Zhou, Mahmut Ünverdi, Allison P Kann, Data curation, Formal analysis; Robert S Krauss, Data curation, Formal analysis, Funding acquisition, Methodology, Writing - review and editing; Atsushi Asakura, Conceptualization, Resources, Data curation, Formal analysis, Supervision, Funding acquisition, Validation, Investigation, Visualization, Methodology, Writing - original draft, Project administration, Writing - review and editing

## Author ORCIDs

Mayank Verma ⓘ http://orcid.org/0000-0003-0167-0842
Yoko Asakura ⓘ https://orcid.org/0000-0003-4107-4236
Allison P Kann ⓘ http://orcid.org/0000-0003-0111-9081
Robert S Krauss ⓘ http://orcid.org/0000-0002-7661-3335
Atsushi Asakura ⓘ http://orcid.org/0000-0001-8078-1027

## Ethics

The animals were housed in an SPF environment and were monitored by the Research Animal Resources (RAR) of the University of Minnesota. All protocols (2204-39969A) were approved by the Institutional Animal Care and Usage Committee (IACUC) of the University of Minnesota and complied with the NIH guidelines for the use of animals in research.

## Decision letter and Author response

Decision letter https://doi.org/10.7554/eLife.73592.sa1
Author response https://doi.org/10.7554/eLife.73592.sa2

# Additional files

## Supplementary files

• Transparent reporting form

• Supplementary file 1. Gene Expression Omnibus (GEO) used for this paper was shown in this table, including their repositories and references.

## Data availability

All the data was obtained from NCBI GEO. Microarrays of mouse MuSCs were obtained from GSE3483 (*Fukada et al., 2007*). scRNA-seq of MuSCs and muscle ECs was performed in this study (GSE129057). scRNA-seq of whole muscle was obtained from GSE143437 (*De Micheli et al., 2020*). Bulk RNA-seq of MuSCs, ECs and single muscle fibers was obtained from GSE108739 (*Verma et al., 2018*) and GSE64379 (*Ryall et al., 2015*). Bulk RNA-seq of TU-tagged RNA of MuSCs was obtained from GSE97399 (*van Velthoven et al., 2017*). Bulk RNA-seq of fixed and unfixed MuSCs was obtained from GSE113631 (*Yue et al., 2020*). Exercise, ALS, DMD, BMD, FSHD GSE3307, Early DMD GSE465, mdx GSE466, GRMD GSE69040, MuSCs GSE15155. All arrays were normalized to their respective controls. All arrays and RNA-seq data are listed in *Supplementary file 1.*

The following dataset was generated:

| Author(s) | Year | Dataset title | Dataset URL | Database and Identifier |
|---|---|---|---|---|
| Verma M, Asakura A | 2020 | Single-cell skeletal muscle satellite cells and endothelial cells during homeostasis and regeneration | https://www.ncbi.nlm.nih.gov/geo/query/acc.cgi?acc=GSE129057 | NCBI Gene Expression Omnibus, GSE129057 |

The following previously published datasets were used:

| Author(s) | Year | Dataset title | Dataset URL | Database and Identifier |
|---|---|---|---|---|
| Cosgrove BD, De Micheli AJ | 2020 | Single-cell transcriptomic atlas of the mouse regenerating muscle tissue | https://www.ncbi.nlm.nih.gov/geo/query/acc.cgi?acc=GSE143437 | NCBI Gene Expression Omnibus, GSE143437 |
| Verma M, Asakura A | 2018 | Skeletal muscle satellite cells, endothelial cells and single muscle fibers | https://www.ncbi.nlm.nih.gov/geo/query/acc.cgi?acc=GSE108739 | NCBI Gene Expression Omnibus, GSE108739 |
| Ryall JG, Dell'Orso S, Derfoul A, Juan A, Zare H, Feng X, Clermont D, Koulnis M, Gutierrez-Cruz G, Fulco M, Sartorelli V | 2015 | The NAD+-Dependent SIRT1 Deacetylase Translates a Metabolic Switch into Regulatory Epigenetics in Skeletal Muscle Stem Cells | https://www.ncbi.nlm.nih.gov/geo/query/acc.cgi?acc=GSE64379 | NCBI Gene Expression Omnibus, GSE64379 |
| Rando TA, van Velthoven CT | 2017 | Transcriptional profiling of quiescent muscle stem cells in vivo | https://www.ncbi.nlm.nih.gov/geo/query/acc.cgi?acc=GSE97399 | NCBI Gene Expression Omnibus, GSE97399 |
| Yue L, Wan R, Cheung TH | 2020 | Transcriptome profiling of quiescent muscle stem cells in vivo | https://www.ncbi.nlm.nih.gov/geo/query/acc.cgi?acc=GSE113631 | NCBI Gene Expression Omnibus, GSE113631 |
| Fukada S | 2007 | Genome-wide expression analysis of satellite cells | https://www.ncbi.nlm.nih.gov/geo/query/acc.cgi?acc=GSE3483 | NCBI Gene Expression Omnibus, GSE3483 |
| Hoffman EP | 2005 | Comparative profiling in 13 muscle disease groups | https://www.ncbi.nlm.nih.gov/geo/query/acc.cgi?acc=GSE3307 | NCBI Gene Expression Omnibus, GSE3307 |
| Chen YW, Zhao P, Borup R, Hoffman EP | 2003 | Expression profiling in the muscular dystrophies | https://www.ncbi.nlm.nih.gov/geo/query/acc.cgi?acc=GSE465 | NCBI Gene Expression Omnibus, GSE465 |
| Tseng BS, Zhao P, Pattison JS, Gordon SE, Granchelli JA, Madsen RW, Folk LC, Hoffman EP, Booth FW | 2003 | mRNA expression in regenerated mdx mouse skeletal muscle | https://www.ncbi.nlm.nih.gov/geo/query/acc.cgi?acc=GSE466 | NCBI Gene Expression Omnibus, GSE466 |
| Moreira YB, Vieira N | 2015 | Duchene Muscular Dystrophy Dogs Escapers and Affected Muscle Dogs Compared to Normal Dogs | https://www.ncbi.nlm.nih.gov/geo/query/acc.cgi?acc=GSE69040 | NCBI Gene Expression Omnibus, GSE69040 |
| Pallafacchina G, Montarras D, Regnault B, Buckingham M | 2010 | Gene profiling of quiescent and activated skeletal muscle satellite cells by an in vivo approach | https://www.ncbi.nlm.nih.gov/geo/query/acc.cgi?acc=GSE15155 | NCBI Gene Expression Omnibus, GSE15155 |

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

## Appendix 1

**Appendix 1—key resources table**

| Reagent type (species) or resource | Designation | Source or reference | Identifiers | Additional information |
|---|---|---|---|---|
| Strain (*Mus musculus*) | Flt1LoxP/LoxP | The Jackson Laboratory | JAX: 028098 | Mouse line obtained from Guo-Hua Fong |
| Strain (*Mus musculus*) | VEGFA+/Hyper | The Jackson Laboratory | JAX: 027314 | |
| Strain (*Mus musculus*) | Pax7CreERT2/+ | The Jackson Laboratory | JAX: 017763 | |
| Strain (*Mus musculus*) | mdx5cv | The Jackson Laboratory | JAX: 002379 | |
| Strain (*Mus musculus*) | Flk1GFP | The Jackson Laboratory | JAX: 017006 | Mouse line obtained from Masatsugu Ema |
| Strain (*Mus musculus*) | R26RtdT | The Jackson Laboratory | JAX: 007909 | |
| Strain (*Mus musculus*) | VEGFALoxP/LoxP | | | Mouse line obtained from Napoleone Ferrara |
| Genetic reagent | Lentiviral pCCL-E4ORF1 | PMID:20972423 | | Viral vector production |
| Genetic reagent | Lentiviral pCCL-myrAkt1 | PMID:20972423 | | Viral vector production |
| Genetic reagent | pCMV-VSV-G | Addgene | 8454 | Viral vector production |
| Genetic reagent | pRSV-Rev | Addgene | 12253 | Viral vector production |
| Genetic reagent | pMDLg/pRRE | Addgene | 12251 | Viral vector production |
| Genetic reagent | RNAscope Probe - Mm-Flt1 (C1) | ACDBio | 415541 | RNAscope |
| Cell line (*Homo sapiens*) | 293 FT | ThermoFisher Scoentific | R70007 | Viral vector production |
| Cell line (*Mus musculus*) | C2C12 | American Type Culture Collection (ATCC) | CRL-1772 | MuSC line |
| Cell line (*Mus musculus*) | bEnd.3 | American Type Culture Collection (ATCC) | CRL-2299 | EC line |
| Cell line (*Mus musculus*) | C166 | American Type Culture Collection (ATCC) | CRL-2581 | EC line |
| Antibody | Anti-CD31-PE (Rat monoclonal) | ThermoFisher Scientific | 12-0311-82; RRID: AB_465632 | 1:200 |
| Antibody | Anti-CD45-PE (Rat monoclonal) | ThermoFisher Scientific | 12-0451-81; RRID: AB_465668 | 1:200 |
| Antibody | Anti-Sca-1-PE (Rat monoclonal) | ThermoFisher Scientific | A18486; RRID: AB_2535332 | 1:200 |
| Antibody | Anti-FLT1 (Mouse monoclonal) | R&D systems | MAB4711; RRID: AB_2107038 | 1:200 |
| Antibody | Anti-FLT1-APC (Rat monoclonal) | R&D systems | FAB4711A; RRID: AB_622149 | 1:200 |
| Antibody | Anti-FLK1 (Rat monoclonal) | BD Biosceinces | 555307; RRID: AB_395720 | 1:200 |
| Antibody | Anti-FLK1-APC (Rat monoclonal) | BD Biosceinces | 560070; RRID: C28AB_1645226 | 1:200 |
| Antibody | Anti-VE-cadherin (Rat monoclonal) | BD Biosciences | 555289; RRID: AB_2244723 | 1:200 |

*Appendix 1 Continued on next page*

*Appendix 1 Continued*

| Reagent type (species) or resource | Designation | Source or reference | Identifiers | Additional information |
|---|---|---|---|---|
| Antibody | Anti-NRP1 (Rabbit monoclonal) | Cell Signaling Technology | 3725; RRID:AB_2155231 | 1:200 |
| Antibody | Anti-NRP1-APC (Rat monoclonal) | R&D systems | FAB5994A | 1:200 |
| Antibody | Anti-NRP2 (Rabbit monoclonal) | Cell Signaling Technology | 3366; RRID: AB_2155250 | 1:200 |
| Antibody | Anti-NRP2-APC (Mouse monoclonal) | R&D systems | FAB22151A; RRID: AB_10973479 | 1:200 |
| Antibody | Anti-Sca-1-PE (Rat monoclonal) | eBiosciences | 12-5981-81; RRID: AB_466085 | 1:200 |
| Antibody | Anti-integrin a7-biotin (Mouse monoclonal) | Miltenyi Biotec | 130-101-979; RRID: AB_2652472 | 1:200 |
| Antibody | Anti-IgG-APC (Mouse) | ThermoFisher Scientific | PA5-33237; RRID: AB_2550652 | 1:200 |
| Antibody | Anti-IgG-APC (Rat) | ThermoFisher Scientific | 17-4321-81; RRID: AB_470181 | 1:200 |
| Antibody | Anti-MyHC (Mouse monoclonal) | Developmental Study Hybridoma Bank | MF-20; RRID: AB_2147781 | 1:50 |
| Antibody | Anti-MyoD (Mouse monoclonal) | DAKO | M3512; RRID: AB_2148874 | 1:200 |
| Antibody | Anti-Pax7 (Mouse monoclonal) | Developmental Study Hybridoma Bank | PAX7; RRID: AB_528428 | 1:20 |
| Antibody | Anti-pAKT (Rabbit monoclonal) | Cell Signaling | 4060; RRID: AB_2315049 | 1:500 |
| Antibody | Anti-VEGFA (Rabbit monoclonal) | Abcam | Ab52917; RRID: AB_883427 | 1:500 |
| Antibody | Anti-mouse IgG Alexa488 | ThermoFisher Scientific | A-11001; RRID: AB_2534069 | 1:1000 |
| Antibody | Anti-rat IgG Alexa488 | ThermoFisher Scientific | A-11006; RRID: AB_141373 | 1:1000 |
| Antibody | Anti-rabbit IgG Alexa488 | ThermoFisher Scientific | A-11008; RRID: AB_143165 | 1:1000 |
| Antibody | Anti-rabbit IgG Alexa568 | ThermoFisher Scientific | A-11011; RRID: AB_143157 | 1:1000 |
| Antibody | Anti-mouse IgG Alexa568 | ThermoFisher Scientific | A-11004; RRID: AB_2534072 | 1:1000 |
| Antibody | Anti-rat IgG Alexa568 | ThermoFisher Scientific | A-11077; RRID: AB_2534121 | 1:1000 |
| Antibody | Anti-biotin beads | Miltenyi Biotec | 130-090-485; RRID: AB_244365 | MACS (1:500) |
| Antibody | Anti-PE beads | Miltenyi Biotec | 130-048-801; RRID: AB_244373 | MACS (1:500) |
| Antibody | Anti-Activated Caspase-3 | Abcam | ab214430; RRID: AB_2938798 | 1:200 |
| Antibody | Anti-phosphoAKT1 | Cell Signaling | 4060; RRID: AB_2315049 | 1:200 for IHC, 1:1000 for WB |
| Antibody | Anti-GAPDH HRP conjugated | Cell Signaling | 3683; RRID: AB_1642205 | 1:10000 |

*Appendix 1 Continued on next page*

*Appendix 1 Continued*

| Reagent type (species) or resource | Designation | Source or reference | Identifiers | Additional information |
|---|---|---|---|---|
| Antibody | Anti-eMHC (Mouse monoclonal) | Developmental Study Hybridoma Bank | F1.652; RRID: AB_528358 | 1:200 |
| Antibody | Anti-Laminin-2 (Rat monoclonal) | MilliporeSigma | L0663; RRID: AB_477153 | 1:500 |
| Sequence-based reagent (Oligo DNA) | TTAAACGAACGTACTTGCAGATG | Integrated DNA Technologies (IDT) | N/A | qPCR Vegfa 92 bp (Forward) |
| Sequence-based reagent (Oligo DNA) | AGAGGTCTGGTTCCCGAAA | Integrated DNA Technologies (IDT) | N/A | qPCR Vegfa 92 bp (Reverse) |
| Sequence-based reagent (Oligo DNA) | GCAGAGCCAGGAACATATACACA | Integrated DNA Technologies (IDT) | N/A | qPCR mFlt1 103 bp(Forward) |
| Sequence-based reagent (Oligo DNA) | GAGATCCGAGAGAAAATGGCCTTT | Integrated DNA Technologies (IDT) | N/A | qPCR mFlt1 103 bp (Reverse) |
| Sequence-based reagent (Oligo DNA) | GCAGAGCCAGGAACATATACACA | Integrated DNA Technologies (IDT) | N/A | qPCR sFlt1 73 bp (Forward) |
| Sequence-based reagent (Oligo DNA) | CAGTGCTCACCTCTAACG | Integrated DNA Technologies (IDT) | N/A | qPCR sFlt1 73 bp (Reverse) |
| Sequence-based reagent (Oligo DNA) | CAGTGGTACTGGCAGCTAGAAG | Integrated DNA Technologies (IDT) | N/A | qPCR KDR/Flk1 66 bp (Forward) |
| Sequence-based reagent (Oligo DNA) | ACAAGCATACGGGCTTGTTT | Integrated DNA Technologies (IDT) | N/A | qPCR KDR/Flk1 66 bp (Reverse) |
| Sequence-based reagent (Oligo DNA) | TCCTGGGAAACTGGTATATCTATGA | Integrated DNA Technologies (IDT) | N/A | qPCR Nrp1 75 bp (Forward) |
| Sequence-based reagent (Oligo DNA) | CATTCCAGAGCAAGGATAATCTG | Integrated DNA Technologies (IDT) | N/A | qPCR Nrp1 75 bp (Reverse) |
| Sequence-based reagent (Oligo DNA) | ATGGCTGGACACCCAATTT | Integrated DNA Technologies (IDT) | N/A | qPCR Nrp2 67 bp (Forward) |
| Sequence-based reagent (Oligo DNA) | ATGGTTAGGAAGCGCAGGT | Integrated DNA Technologies (IDT) | N/A | qPCR Nrp2 67 bp (Reverse) |
| Sequence-based reagent (Oligo DNA) | CACCTGGAGAGGATGAAGAAGAA | Integrated DNA Technologies (IDT) | N/A | qPCR Myh3 298 bp (Forward) |
| Sequence-based reagent (Oligo DNA) | AAGACTTGACTTTCACTTGGAGTTTA TC | Integrated DNA Technologies (IDT) | N/A | qPCR Myh3 298 bp (Reverse) |
| Sequence-based reagent (Oligo DNA) | TTCGGAAGCTCCTTCTGTTT | Integrated DNA Technologies (IDT) | N/A | qPCR Htatsf1 79 bp (Forward) |
| Sequence-based reagent (Oligo DNA) | CCAGAGTCTGAATACAATGGTCA | Integrated DNA Technologies (IDT) | N/A | qPCR Htatsf1 79 bp (Reverse) |
| Sequence-based reagent (Oligo DNA) | CGCACGGCCGGTACAGTGAAACTG | Integrated DNA Technologies (IDT) | N/A | qPCR 18 S rRNA 343 bp (Forward) |

*Appendix 1 Continued on next page*

*Appendix 1 Continued*

| Reagent type (species) or resource | Designation | Source or reference | Identifiers | Additional information |
|---|---|---|---|---|
| Sequence-based reagent (Oligo DNA) | CGCACGGCCGGTACAGTGAA ACTG | Integrated DNA Technologies (IDT) | N/A | qPCR 18 S rRNA 343 bp (Reverse) |
| Sequence-based reagent (Oligo DNA) | CTTGCTCACCATGGTCAGCT GCTG | Integrated DNA Technologies (IDT) | N/A | Genotyping Myh3 Exon1 WT/ MUT-103bp (Reverse) |
| Sequence-based reagent (Oligo DNA) | CACTTTTAACTTCGACCCTGAGCC | Integrated DNA Technologies (IDT) | N/A | Genotyping Myh3 Exon2 WT/ MUT-103bp (Reverse) |
| Sequence-based reagent (Oligo DNA) | GGCCAGACTCTCTTTCTCAA GTGC | Integrated DNA Technologies (IDT) | N/A | Genotyping Myh3 Exon2 WT-135bp (Forward) |
| Sequence-based reagent (Oligo DNA) | GCAGAATTGCCTGTTATCCCTCCC | Integrated DNA Technologies (IDT) | N/A | Genotyping Myh3 Exon3 WT-135bp (Reverse) |
| Sequence-based reagent (Oligo DNA) | GCTGCTGTTGATTACCTGGC | Integrated DNA Technologies (IDT) | N/A | Genotyping Pax7CreERT2 (Common) |
| Sequence-based reagent (Oligo DNA) | CAAAAGACGGCAATATGGTG | Integrated DNA Technologies (IDT) | N/A | Genotying Pax7CreERT2 MUT-235bp (Reverse) |
| Sequence-based reagent (Oligo DNA) | CTGCACTGAGACAGGACCG | Integrated DNA Technologies (IDT) | N/A | Genotying Pax7CreERT2 WT-419bp (Reverse) |
| Sequence-based reagent (Oligo DNA) | GGCATTAAAGCAGCGTATCC | Integrated DNA Technologies (IDT) | N/A | Genotyping R26RtdT (9103) MUT-196bp (Forward) |
| Sequence-based reagent (Oligo DNA) | CTGTTCCTGTACGGCATGG | Integrated DNA Technologies (IDT) | N/A | Genotyping R26RtdT (9105) MUT-196bp (Reverse) |
| Sequence-based reagent (Oligo DNA) | AAGGGAGCTGCAGTGGAGTA | Integrated DNA Technologies (IDT) | N/A | Genotyping R26RtdT (9020) WT-297bp (Forward) |
| Sequence-based reagent (Oligo DNA) | CCGAAAATCTGTGGGAAGTC | Integrated DNA Technologies (IDT) | N/A | Genotyping R26RtdT (9021) WT-297bp (Reserve) |
| Sequence-based reagent (Oligo DNA) | AGCAGCACGACTTCTTCAAG TCCG | Integrated DNA Technologies (IDT) | N/A | Genotying Flk1GFP-161bp (Forward) |
| Sequence-based reagent (Oligo DNA) | CTCCTTGAAGTCGATGCCCT TCAG | Integrated DNA Technologies (IDT) | N/A | Genotying Flk1GFP-161bp (Reserve) |
| Sequence-based reagent (Oligo DNA) | CGCTTTTTGTCAGTCATCTTCA | Integrated DNA Technologies (IDT) | N/A | Genotying Flt1Loxp/Loxp (FlpeEX3F) WT-223bp/MUT-641bp (Forward) |
| Sequence-based reagent (Oligo DNA) | GTGCCACTGACCTAACATGT AAGAG | Integrated DNA Technologies (IDT) | N/A | Genotying Flt1Loxp/Loxp (FlpeInt3R) WT-223bp/MUT-641bp (Reverse) |
| Sequence-based reagent (Oligo DNA) | CCA TAG ATG TGA CAA GCC AAG | Integrated DNA Technologies (IDT) | N/A | Genotyping VEGFAHyper (24286) WT-254bp (Forward) |
| Sequence-based reagent (Oligo DNA) | ACC CGG GGA TCC TCT AGA AC | Integrated DNA Technologies (IDT) | N/A | Genotyping VEGFAHyper (25307) MUT-199bp (Forward) |

*Appendix 1 Continued on next page*

*Appendix 1 Continued*

| Reagent type (species) or resource | Designation | Source or reference | Identifiers | Additional information |
|---|---|---|---|---|
| Sequence-based reagent (Oligo DNA) | GAC CGT GCT TGG TCA CCT | Integrated DNA Technologies (IDT) | N/A | Genotyping VEGFAHyper (25308, Common) WT-254bp/ MUT-199bp (Reverse) |
| Sequence-based reagent (Oligo DNA) | CCTGGCCCTCAAGTACACCTT | Integrated DNA Technologies (IDT) | N/A | VEGFALoxP/LoxP (muVEGF 419 .F) WT-106bp/MUT-148bp (Forward) |
| Sequence-based reagent (Oligo DNA) | TCCGTACGACGCATTTCTAG | Integrated DNA Technologies (IDT) | N/A | VEGFALoxP/LoxP (muVEGF 567 .R) WT-106bp/MUT-148bp (Reverse) |
| Sequence-based reagent (Oligo DNA) | GAAGCTCCCAGAGACAAGTC | Integrated DNA Technologies (IDT) | N/A | mdx5cv (0981) WT/MUT-180bp (Forward) |
| Sequence-based reagent (Oligo DNA) | TCATGAGCATGAAACTGTTCTT | Integrated DNA Technologies (IDT) | N/A | mdx5cv (0981) WT/MUT-180bp (Reverse) |
| Commercial assay or kit | GoTaq qPCR Master Mix | Promega, | A6002 | qPCR |
| Commercial assay or kit | Click-iT EdU Alexa Fluor 488 Imaging Kit | Thermo Scientific | C10337 | EdU staining |
| Commercial assay or kit | Crystal violet Assay Kit | Abcam | ab232855 | Cell viability assay |
| Commercial assay or kit | Transcriptor First Strand cDNA Synthesis Kit | Roche-Sigma-Aldrich | 4379012001 | cDNA synthesis |
| Commercial assay or kit | Annexin V-Biotin Apoptosis Detection Kit | eBioscience | BMS500BT-100 | Appotosis assay |
| Commercial assay or kit | DirectZolTM RNA Microprep Kit | Zymo Research | R2062 | RNA isolation |
| Commercial assay or kit | Midi Fast Ion Plasmid Kit | IBI Scientific | IB47111 | Plasmid DNA isolation |
| Commercial assay or kit | H&E Staining Kit | Abcam | ab245880 | Histology |
| Commercial assay or kit | SuperSignal West Femto chemiluminescent substrate | Fisher Scientific | PI37074 | Western Blotting |
| Commercial assay or kit | NE-PER Nuclear and Cytoplasmic Extraction reagents | Thermo Fisher Scientific | 78833 | Protein extraction |
| Chemical compound, drug | bFGF | Thermo Fisher Scientific | PHG0263 | MuSC culture (20 ng/ml) |
| Chemical compound, drug | Bovine serum albumin (BSA) | Jackson Immuno Research | 10001620 | FACS/Immunostaining (1–2%) |
| Chemical compound, drug | Chicken embryo extract | MP-Biomedical | 92850145 | Single muscle fiber culture (0.5%) |
| Chemical compound, drug | Collagen | BD Biosciences | 354236 | Culture dish coating (0.01%) |
| Chemical compound, drug | Collagenase type I | Sigma-Aldrich | C0130 | Single myofiber isolation (0.2%) |
| Chemical compound, drug | Collagenase type II | Worthington Biochemical Corp | CLD-2 | FACS and MuSC/ES isolation (0.2%) |
| Chemical compound, drug | DAPI (4',6-diamidino-2-phenylindole dihydrochloride) | Thermo Fisher Scientific | D1306 | DNA staining (1 µg/ml) |
| Chemical compound, drug | Dulbecco's Modified Eagle's Medium (DMEM) | Thermo Fisher Scientific | 41966 | Single myofiber and ES culture |

*Appendix 1 Continued on next page*

*Appendix 1 Continued*

| Reagent type (species) or resource | Designation | Source or reference | Identifiers | Additional information |
|---|---|---|---|---|
| Chemical compound, drug | EdU | Thermo Fisher Scientific | C10340 | Cell proliferation (1 µM) |
| Chemical compound, drug | F-10 Ham's media | Sigma-Aldrich | N6635 | MuSC culture |
| Chemical compound, drug | Fetal calf serum (FCS) | Atlas Biological | FS-0500-AD | MuSC/EC/Single muscle fiber culture (10% or 20%) |
| Chemical compound, drug | Horse serum | Thermo Fisher Scientific | 26050088 | 2% (coating), 5% (Differentiation culture) |
| Chemical compound, drug | L-glutamine | Thermo Fisher Scientific | 25030 | Culture medium (20 mM) |
| Chemical compound, drug | Matrigel | Corning Life Sciences | 354230 | Single muscle fiber culture (1:20) |
| Chemical compound, drug | Penicillin/streptomycin | Life Technologies | 15140 | Culture medium (1 X) |
| Chemical compound, drug | Tamoxifen (TMX) | Sigma-Aldrich | T5648 | Cre recombinase (60 mg/kg i.p.) |
| Chemical compound, drug | 4-hydroxy tamoxifen (4-OHT) | Sigma-Aldrich | H6278 | Culture (1 µM) |
| Chemical compound, drug | Cardiotoxin (CTX) | Sigma-Aldrich | V9125 | Muscle injury (10 µM) |
| Chemical compound, drug | DraIII | New England Biolabs | R3510S | DNA digestion (500 U/ml) |
| Chemical compound, drug | ZombieNIR | Biolegends | 423105 | FACS (0.1%) |
| Chemical compound, drug | recombinant VEGFA | R&D Systems | 493 MV | Culture (2–100 ng/ml) |
| Chemical compound, drug | Thapsigargin | Sigma-Aldrich | T9033 | Culture (1 µM) |
| Chemical compound, drug | FLT1-FC | R&D Systems | 7756-FL | Culture (100 ng/ml) |
| Chemical compound, drug | ZM306416 | R&D Systems | 2499/1 | Culture (3 µM) |
| Chemical compound, drug | SU5402 | R&D Systems | 3300/1 | Culture (10 µM) |
| Chemical compound, drug | EG00229 | R&D Systems | 6986/10 | Culture (30 µM) |
| Chemical compound, drug | PolyJet transfection reagent | Signagen Laboratories | SL100688 | DNA transfection (10 µl for 6 cm plate) |
| Chemical compound, drug | Polybrene | MilliporeSigma | H9268 | Viral infection (0.8 µg/ml) |
| Chemical compound, drug | Saponin | ThermoFisher Scientific | ICN10285525 | Immunostaining (0.01%) |
| Chemical compound, drug | Direct Red 80 | Sigma-Aldrich | 365548 | Histology (0.1%) |
| Chemical compound, drug | Proteinase K | Sigma-Aldrich | P2308 | DNA isolation (40 µg/mL) |
| Chemical compound, drug | Oil Red O solution | Sigma-Aldrich | O1391-250ML | Histology (0.5%) |
| Chemical compound, drug | Trizol | ThermoFisher Scientific | 15596026 | RNA isolation |

*Appendix 1 Continued on next page*

*Appendix 1 Continued*

| Reagent type (species) or resource | Designation | Source or reference | Identifiers | Additional information |
|---|---|---|---|---|
| Software, algorithm | Photoshop 2020 | Adobe | https://www.adobe.com/products/photoshop.html | Imaging analysis |
| Software, algorithm | Fiji | NIH | https://imagej.net/software/fiji/ | Imaging analysis |
| Software, algorithm | cellSens Entry 1.11 | Olympus | https://www.olympus-lifescience.com/en/software/cellsens/ | Microscopy |
| Software, algorithm | Prism 9 | GraphPad | https://www.graphpad.com/support/faq/prism-900-release-notes/ | Data analysis and statistics |
| Software, algorithm | RStudio | RStudio | https://www.rstudio.com | Data analysis and statistics |
| Software, algorithm | Autodesk Graphic | Autodesk | https://www.graphic.com | Vector design |
| Software, algorithm | BioRender: Scientific Image and Illustration Software | BioRender.com | https://www.biorender.com | Illustrator |
| Other | Anti-biotin beads | Miltenyi Biotec | 130-090-485; RRID:AB_244365 | MACS |
| Other | Anti-PE beads | Miltenyi Biotec | 130-048-801; RRID:AB_244373 | MACS |
| Other | Cell culture plate, 24 well | Sarstedt | 83.3922 | Single myofiber culture |
| Other | Tissue culture dish | Sarstedt | 83.39 | MuSC/ES culture |
| Other | Tissue culture dish | Sarstedt | 83.3901 | MuSC/ES culture |
| Other | Tissue culture dish | Sarstedt | 83.3902 | MuSC/ES culture |
| Other | LD column | Miltenyi Biotec | 130-042-901 | MACS |
| Other | MS column | Miltenyi Biotec | 130-042-201 | MACS |
| Other | Fortessa X-20 | BD Biosciences | | FACS |
| Other | BD FACSAria | BD Biosciences | | FACS |
| Other | moorLabTM laser Doppler | Moor Instruments | MOORVMS-LDF | Laser Doppler flow |
| Other | Grip strength meter | Columbus Instruments | 1027CSM-D54 | Forelimb muscle force |
| Other | iBright FL1500 | ThermoFisher Scientific | A44115 | Western blotting |
| Other | Olympus IX81 Inverted Fluorescense microscope | Olympus | | Microscope |
| Other | Olympus BX51 Fluorescense microscope | Olympus | | Microscope |

