## [Editor Report]

This study presents a valuable finding on the unique role of VEGFA-FLT1-AKT1 signaling in regulating muscle stem cell (MuSC) survival. The evidence supporting the claims is convincing, with multiple approaches utilized, including pharmacological and genetic methods performed in vitro and in vivo, demonstrating that the VEGFA-FLT1-AKT1 axis protected MuSCs from apoptosis. The work will be of broad interest to researchers in the MuSC biology field and support the future development of VEGFA and FLT1 targeted therapies for various diseases, such as cancer and neuromuscular diseases.

---

## [Decision Letter]

**Decision letter after peer review:**

Thank you for submitting your article "Endothelial cell signature in muscle stem cells validated by VEGFA-FLT1-AKT1 axis promoting survival of muscle stem cell" for consideration by *eLife*. Your article has been reviewed by 3 peer reviewers, and the evaluation has been overseen by a Reviewing Editor and Kathryn Cheah as the Senior Editor. The following individuals involved in review of your submission have agreed to reveal their identity: Francesco Saverio Tedesco (Reviewer #2); Shihuan Kuang (Reviewer #4).

Essential revisions (for the authors):

(1) Using exogenous and blocking VEGFA in vitro improves or inhibits survival, respectively, and in vivo experiments support the notion of cell- autonomous function of VEGFA in SCs. Nevertheless, it is intriguing that VEGFA had no effects on cell survival in response to ER-stress. Did the authors test different concentrations of VEGFA or test if VEGFA in the culture serum or secreted from cultured myoblasts is sufficient to protect ER-stress induced apoptosis?

(2) Some discussion points should be addressed. First, the potential redundant and distinct functions of signaling through FLT1 and FLK1 should be addressed. Is VEGFA signaling through these receptors dose dependent or dependent on the co-receptors? What is the effect of exogenous VEGFA on VEGFA-KO SCs? Can the authors add VEGFA in vitro to the KO SCs to determine if there is a threshold level that FLT1-decoy mediates that may be important to exert the cell-autonomous effect.

(3) The authors also analyzed publicly available datasets to show that Vegfa expression is reduced in skeletal muscles of DMD patients and animal models. Furthermore, deletion of Flt1 in mdx aggravates, whereas Vegfa overexpression ameliorates muscle pathology. This conclusion could be consolidated by explaining or investigating why Vegfa expression is reduced in dystrophic muscle? For example, is it due reduced MuSCs as they are the main source of VEGFA? Please expand the discussion on whether the improved phenotype is mainly due to improved satellite cell survival or due to improved vascular differentiation or function in response to Vegfa overexpression. Is there any evidence that satellite cell survival is impaired in mice with myopathies, and if survival is improved in the VEGFA+/Hyper/mdx mice relative to mdx mice? Please provide some evidence to support that (i) satellite cell survival is impaired in mdx mice, and (ii) activation of VEGFA signaling improves survival of satellite cells in the mdx mice. This in vivo data would better justify the main conclusion of the study.

(4) Figure 2H: phospho-Akt should be quantified via a more quantitative method such as western blot.

(5) Figure 3 While in the manuscript muscle, it is claimed that muscle injury has been induced by BaCl2, in the figure it is mentioned that muscle injury has been induced by CTX. Please clarify.

(6) Figure 3B: The representative HandE staining pictures shown for VEGFA+/+ and VEGFA+/hyper show inflammatory cell infiltration (accumulation of nuclei between myocytes). What about muscle inflammation in your models. Inflammatory cells are also a big source of VEGFA in injured skeletal muscle.

(7) Figure 3D How are Musc quantified?

8) Figure 3E, F How is ferret diameter measured?

(9) Figure 3G How is Anexin 5 staining quantified? Do the histograms show total Anexin 5+ cells or only double Pax7+/Anexin 5+ cells?

(10) Figure 4, the authors should quantify capillary density to strengthen their laser blood flow data.

(11) Figure 4G and 4J legends and figure panels do not seem to correspond.

(12) Since the authors used Pax7tdT mice they should be able to track the fate of VegfA+/Hyper, VegfaKO or Flt1KO satelite cells upon injury and to quantify the number and/or percentage of tomato+ myocytes after injury in each model reflecting the specific consequences of impaired VEGFA signaling in satellite cells.

(13) Supp Figure 3 How was muscle injury induced what injury

(14) Sup Figure 3F is poorly representative, myocyte seem to be larger in MuSC-VEGFA KO mice mice.

(15) Line 180-182: VEGFA is repeated, and the sentence should be rewritten.

(16) Standard convention for protein and gene name should be used throughout the study. For example, VEGFA should be used for the protein and Vegfa (italic) should be used for the gene and mRNA.

---

## [Author Response]

Essential revisions (for the authors):(1) Using exogenous and blocking VEGFA in vitro improves or inhibits survival, respectively, and in vivo experiments support the notion of cell- autonomous function of VEGFA in SCs. Nevertheless, it is intriguing that VEGFA had no effects on cell survival in response to ER-stress. Did the authors test different concentrations of VEGFA or test if VEGFA in the culture serum or secreted from cultured myoblasts is sufficient to protect ER-stress induced apoptosis?

Thanks for pointing out that this aspect needed to be clarified. VEGFA in fetal bovine serum (FBS) is expected to be in the femtogram range, so we do not expect it to be a significant contributing factor. Since myoblasts themselves secrete a high amount of VEGFA, we think that endogenous VEGFA may be responsible for the lack of exogenous effect of VEGFA in the absence of other inhibitors. There are reports of intracellular VEGFA having a distinct role in cell survival (Lee et al., 2007). Our data further supports this, which shows that *Vegfa*-KO MuSCs are exquisitely sensitive to apoptosis after injury (Figure 3G). In addition, we tried a dose escalation of VEGFA. We did see improved cell survival in myoblast following UV light-induced apoptosis at 20 ng/ml compared to 50 ng/ml (Figure 2—figure supplement 1E and 1F). This makes biological sense as the concentration of VEGFA has distinct downstream effects (Noren et al., 2016).

Lee S, Chen TT, Barber CL, Jordan MC, Murdock J, Desai S, Ferrara N, Nagy A, Roos KP, Iruela-Arispe ML. 2007. Autocrine VEGF signaling is required for vascular homeostasis. *Cell* 130:691–703. doi:10.1016/j.cell.2007.06.054

Noren DP, Chou WH, Lee SH, Qutub AA, Warmflash A, Wagner DS, Popel AS, Levchenko A. 2016. Endothelial cells decode VEGF-mediated ca^2+^ signaling patterns to produce distinct functional responses. *Sci Signal* 9:ra20–ra20. doi:10.1126/scisignal.aad3188

(2) Some discussion points should be addressed. First, the potential redundant and distinct functions of signaling through FLT1 and FLK1 should be addressed. Is VEGFA signaling through these receptors dose dependent or dependent on the co-receptors? What is the effect of exogenous VEGFA on VEGFA-KO SCs? Can the authors add VEGFA in vitro to the KO SCs to determine if there is a threshold level that FLT1-decoy mediates that may be important to exert the cell-autonomous effect.

The differential functions of VEGFA on FLT1 and FLK1 in MuSC are indeed exciting. During the review of this paper, two groups have either published (Groppa et al., 2023) or submitted a pre-print (Chen et al., 2022) complementing this work. While FLK1 was not the focus of our report, both these reports have performed extensive experiments on the role of FLK1 in proliferation. The discussion regarding these reports and how they complement our report has been added to the Discussion section. The endogenous vs exogenous VEGFA is a complicated system currently under investigation but is outside the scope of this current report.

Regarding co-receptors' effect, KDR (FLK1) works with its co-receptor NRP1. We investigated the inhibition of NRP1 using the pharmacological inhibitor EG00229, which is an NRP1 receptor antagonist for VEGFA and does not affect VEGFA binding to FLT1 of KDR1. We found no effect of NRP1 inhibition on cell survival in vitro. This experiment can be found in Figure 2C and 2D.

Chen W, Wang YX, Ritso M, Perkins TJ, Rudnicki MA. 2022. KDR Signaling in Muscle Stem Cells Promotes Asymmetric Division and Progenitor Generation for Efficient Regeneration. bioRxiv. 2022.06.27.497734. doi: https://doi.org/10.1101/2022.06.27.497734

Groppa E, Martini P, Derakhshan N, Theret M, Ritso M, Tung LW, Wang YX, Soliman H, Hamer MS, Stankiewicz L, Eisner C, Erwan LN, Chang C, Yi L, Yuan JH, Kong S, Weng C, Adams J, Chang L, Peng A, Blau HM, Romualdi C, Rossi FMV. 2023. Spatial compartmentalization of signaling imparts source-specific functions on secreted factors. *Cell Rep.* 42:112051. doi: 10.1016/j.celrep.2023.112051

(3) The authors also analyzed publicly available datasets to show that Vegfa expression is reduced in skeletal muscles of DMD patients and animal models. Furthermore, deletion of Flt1 in mdx aggravates, whereas Vegfa overexpression ameliorates muscle pathology. This conclusion could be consolidated by explaining or investigating why Vegfa expression is reduced in dystrophic muscle? For example, is it due reduced MuSCs as they are the main source of VEGFA? Please expand the discussion on whether the improved phenotype is mainly due to improved satellite cell survival or due to improved vascular differentiation or function in response to Vegfa overexpression. Is there any evidence that satellite cell survival is impaired in mice with myopathies, and if survival is improved in the VEGFA+/Hyper/mdx mice relative to mdx mice? Please provide some evidence to support that (i) satellite cell survival is impaired in mdx mice, and (ii) activation of VEGFA signaling improves survival of satellite cells in the mdx mice. This in vivo data would better justify the main conclusion of the study.

We divided comment five into individual portions to better address them as below.

“The authors also analyzed publicly available datasets to show that Vegfa expression is reduced in skeletal muscles of DMD patients and animal models. Furthermore, deletion of Flt1 in mdx aggravates, whereas Vegfa overexpression ameliorates muscle pathology.”

We have investigated several aspects of FLT1 biology in the *mdx* mice, so we would like to clarify the point about the deletion of *Flt1* in *mdx* mice. While complete knockout of *Flt1* is embryonic lethal, haploinsufficient *Flt1* (*Flt1* heterozygous KO) in the *mdx* background rescues the dystrophic pathology via increased vascular density since FLT1 acts as a decoy receptor and a sink trap for VEGF thereby, preventing excessive normal and pathological angiogenesis (Verma et al., 2010). This can be phenocopied by conditional deletion of *Flt1* in the endothelial cells and with drugs targeting Flt1 (Verma et al., 2019; Bosco et al., 2021). Deletion of *Flt1* in MuSCs does worsen the phenotype in the *mdx* mice. This is an example of the complicated cell type-specific response that can interplay in muscle regeneration concerning VEGFA and its receptors.

Verma M, Asakura Y, Hirai H, Watanabe S, Tastad C, Fong G-H, Ema M, Call JA, Lowe DA, Asakura A. 2010. Flt-1 haploinsufficiency ameliorates muscular dystrophy phenotype by developmentally increased vasculature in mdx mice. Hum Mol Genet 19:4145–59. doi:10.1093/hmg/ddq334

Verma M, Shimizu-Motohashi Y, Asakura Y, Ennen JP, Bosco J, Zhou Z, Fong G, Josiah S, Keefe D, Asakura A. 2019. Inhibition of FLT1 ameliorates muscular dystrophy phenotype by increased vasculature in a mouse model of Duchenne muscular dystrophy. PLOS Genet 15:e1008468. doi:10.1371/journal.pgen.1008468

Bosco J, Zhou Z, Gabriëls S, Verma M, Liu N, Miller BK, Gu S, Lundberg DM, Huang Y, Brown E, Josiah S, Meiyappan M, Traylor MJ, Chen N, Asakura A, De Jonge N, Blanchetot C, de Haard H, Duffy HS, Keefe D. 2021. VEGFR-1/Flt-1 inhibition increases angiogenesis and improves muscle function in a mouse model of Duchenne muscular dystrophy.

“This conclusion could be consolidated by explaining or investigating why Vegfa expression is reduced in dystrophic muscle? For example, is it due reduced MuSCs as they are the main source of VEGFA?”

While MuSCs are a large contributor to VEGFA by cell volume (Verma et al., 2018), the muscle fiber is expected to secrete the most VEGFA, given its disproportionately large volume (Mac Gabhann et al., 2011). We suspect that the VEGFA is reduced in the progressive dystrophic muscle due to these diseases' loss of muscle mass. In addition, the reported RNA-seq data also confirmed the down-regulation of *Vegfa* in DMD and BMD patients, as well as in DMD model dogs and mice, as shown in Figure 4—figure supplement 1A.

Verma M, Asakura Y, Murakonda BSR, Pengo T, Latroche C, Chazaud B, McLoon LK, Asakura A. 2018. Muscle Satellite Cell Cross-Talk with a Vascular Niche Maintains Quiescence via VEGF and Notch Signaling. Cell Stem Cell 23:530-543.e9. doi:10.1016/j.stem.2018.09.007

Mac Gabhann F, Qutub AA, Annex BH, Popel AS. 2011. Systems biology of proangiogenic therapies targeting the VEGF system. Wiley Interdiscip Rev Syst Biol Med 2:694–707. doi:10.1002/wsbm.92

“Please expand the discussion on whether the improved phenotype is mainly due to improved satellite cell survival or due to improved vascular differentiation or function in response to Vegfa overexpression. Is there any evidence that satellite cell survival is impaired in mice with myopathies, and if survival is improved in the VEGFA+/Hyper/mdx mice relative to mdx mice?”

Our goal with the current project was to answer the question stated above whether the improved pathology seen in the *mdx* mouse by indirect increase in VEGFA (via modulation of FLT1) was due to its effect on the vasculature or the MuSC (Verma et al., 2010; Verma et al., 2019; Bosco et al., 2021). By deleting the *Flt1* in the MuSC specifically, we were expecting to decouple the vascular contribution from the MuSC contribution. However, as shown in Figure 4B and Supplemental Figure 4-1D, conditional knockout of *Flt1* in MuSC decreases the capillary density, thus making it difficult to delineate between the two mechanisms of action with the current set of experiments.

“Please provide some evidence to support that (i) satellite cell survival is impaired in mdx mice”

evidence that MuSC survival is impaired comes indirectly from transplantation experiments where satellite cells obtained from *mdx* mice engraft much less than those from WT mice (Boldrin et al., 2015). This sentence was included in the Discussion section.

" (ii) activation of VEGFA signaling improves survival of satellite cells in the mdx mice”

Thank you for recommending this experiment. We performed the specific experiment that was requested. We used the Pax7 lineage tdTomato reporter mice crossed into the mdx mice and the *Vegfa^+/^*^Hyper^ allele to yield *mdx:Vegfa^+/+^:Pax7^tdT^* and *mdx:Vegfa^+/^*^Hyper^*:Pax7^tdT^* mice. We injected BaCl_2_ to create acute TA muscle injury, and the muscle section was stained for activated Caspase-3 (Cas-3) as a marker for apoptotic cell death 72 hrs after the injury. We detected decreased Cas-3(+)tdTomato(+) apoptotic MuSCs in the *Vegfa^+/^*^Hyper^ allele compared with *Vegfa^+/+^* allele, indicating that increased VEGFA improves MuSC survival in the *mdx* mice in vivo. These data are now in the new Figure 4—figure supplement 2A-2C. A similar experiment was performed by Gropa et al. during the review process (Groppa et al., 2023). They evaluated TUNEL(+)MyoD(+) cells in muscle after NTX injury and found them to be elevated in the MuSC*Vegfa*-cKO muscle. Lastly, this data also comes from extrapolation of data from MuSC transplantation studies where VEGFA over-expression in either the MuSC or the host muscle improved the survival of the transplanted cells (Bouchentouf et al., 2008).

Bouchentouf M, Benabdallah BF, Bigey P, Yau TM, Scherman D, Tremblay JP. 2008. Vascular endothelial growth factor reduced hypoxia-induced death of human myoblasts and improved their engraftment in mouse muscles. Gene Ther 15:404–414.

Groppa E, Martini P, Derakhshan N, Theret M, Ritso M, Tung LW, Wang YX, Soliman H, Hamer MS, Stankiewicz L, Eisner C, Erwan LN, Chang C, Yi L, Yuan JH, Kong S, Weng C, Adams J, Chang L, Peng A, Blau HM, Romualdi C, Rossi FMV. 2023. Spatial compartmentalization of signaling imparts source-specific functions on secreted factors. Cell Rep. 42:112051. doi: 10.1016/j.celrep.2023.112051

(4) Figure 2H: phospho-Akt should be quantified via a more quantitative method such as western blot.

We performed the western blotting and quantification as recommended, and the data is presented in new Figure 2—figure supplement 1N and 1O using an Imaging analyzer (iBright FL1500, ThermoFisher Scientific).

(5) Figure 3 While in the manuscript muscle, it is claimed that muscle injury has been induced by BaCl2, in the figure it is mentioned that muscle injury has been induced by CTX. Please clarify

Thank you for this indication. In the experiments shown in Figure 1, we utilized CTX to induce muscle injury, while in the experiments shown in Figure 3, we utilized BaCl_2_ to induce muscle injury. We clarified the methods used for inducing muscle injury in the main text and figure legends.

(6) Figure 3B: The representative HandE staining pictures shown for VEGFA+/+ and VEGFA+/hyper show inflammatory cell infiltration (accumulation of nuclei between myocytes). What about muscle inflammation in your models. Inflammatory cells are also a big source of VEGFA in injured skeletal muscle.

We agree that inflammatory cells are a significant source of VEGFA, and this makes this biology complicated and out of the scope of the current report. However, this was recently touched on in a complimentary publication (Groppa et al., 2023), where they evaluated the expression of VEGFA in inflammatory at different times and also performed lineage-based deletion of *VEGFA* in all hematopoietic cells using the *VAV-Cre* mice. While the data was not shown in the paper, they mentioned that they found a reduction in vessel density and disrupted muscle regeneration in this mouse model.

Groppa E, Martini P, Derakhshan N, Theret M, Ritso M, Tung LW, Wang YX, Soliman H, Hamer MS, Stankiewicz L, Eisner C, Erwan LN, Chang C, Yi L, Yuan JH, Kong S, Weng C, Adams J, Chang L, Peng A, Blau HM, Romualdi C, Rossi FMV. 2023. Spatial compartmentalization of signaling imparts source-specific functions on secreted factors. Cell Rep. 42:112051. doi: 10.1016/j.celrep.2023.112051

(7) Figure 3D How are Musc quantified?

MuSCs are quantified by Pax7 immunofluorescence after single muscle fiber isolation, as reflected in the Figure 3 legend.

(8) Figure 3E, F How is ferret diameter measured?

Feret’s diameter was measured manually using ImageJ (FIJI), as mentioned in the Materials and methods section.

(9) Figure 3G How is Anexin 5 staining quantified? Do the histograms show total Anexin 5+ cells or only double Pax7+/Anexin 5+ cells?

We utilized FACS analysis to quantify Annexin V(+/-) MuSCs. This histogram in Figure 3G shows the populations of Pax7-tdTomato(+)Annexin V(-) live MuSCs and Pax7tdTomato(+)Annexin V(+) apoptotic MuSCs.

(10) Figure 4, the authors should quantify capillary density to strengthen their laser blood flow data.

The CD31(+) capillary densities from the *mdx:MuSC-Flt1^Δ/Δ^* and *mdx:Vegfa^+/^*^Hyper^ mice have been performed and are shown in the new Figure 4B and 4G, and Figure 4—figure supplement 1D. As a result, the capillary densities in the *mdx:*MuSC*-Flt1^Δ/Δ^* mice and *mdx:Vegfa^+/^*^Hyper^ mice are decreased and increased, respectively. This is consistent with the laser Doppler flow data for the *mdx:*MuSC*-Flt1^Δ/Δ^* mice (decreased; Figure 4E) and *mdx:Vegfa^+/^*^Hyper^ mice (increased; Figure 4J).

(11) Figure 4G and 4J legends and figure panels do not seem to correspond

Thank you for bringing this to our attention. We have changed both figure legends.

(12) Since the authors used Pax7tdT mice they should be able to track the fate of VegfA+/Hyper, VegfaKO or Flt1KO satelite cells upon injury and to quantify the number and/or percentage of tomato+ myocytes after injury in each model reflecting the specific consequences of impaired VEGFA signaling in satellite cells.

When the *Pax7^tdT^* mouse muscle is injured, we see a unanimous expression of tdTomato(+) in the muscle fibers in the damaged area, indicating that each muscle fiber that was injured had some contribution from the tdTomato(+) MuSC. We attempted to get around this predicament by looking at the muscle four days after injury. At this point, the muscle fibers are in the active phase of growth after injury, and the size of the muscle fiber could be used as a proxy for the amount of MuSC contribution. We quantified the maturity of the embryonic myosin heavy chain (eMHC)(+) fibers in MuSC*-Vegfa^Δ/Δ^*, MuSC*-Flt1^Δ/Δ^,* and *Vefgfa^+/^*^Hyper^ mice during regeneration. We noticed that in VEGFA+/Hyper mice, the eMHC(+) fibers were significantly more prominent, and reciprocally, the eMHC+ fibers in the MuSC*-Vegfa^Δ/Δ^* and MuSC*-Flt1^Δ/Δ^* mice were smaller (new Figure 3—figure supplement 1E and 1F). Since the fibers were not different in size at baseline (Figure 3B, 3F, 3J, and 3N), we can expect these changes to result from the MuSC contribution during regeneration.

(13) Supp Figure 3 How was muscle injury induced what injury.

Muscle injury was induced by intramuscular injection of BaCl_2_, as listed in Figure 3A.

(14) Sup Figure 3F is poorly representative, myocyte seem to be larger in MuSC-VEGFA KO mice mice.

Thank you for highlighting this. We have now put in more representative images in Figure 3—figure supplement 2B.

(15) Line 180-182: VEGFA is repeated, and the sentence should be rewritten.

We rewrote the sentence.

(16) Standard convention for protein and gene name should be used throughout the study. For example, VEGFA should be used for the protein and Vegfa (italic) should be used for the gene and mRNA.

Thank you for pointing this out. We have made the changes accordingly throughout the manuscript.